# Differential expression of Lutheran/BCAM regulates biliary tissue remodeling in ductular reaction during liver regeneration

Yasushi Miura[1,2], Satoshi Matsui[1,3], Naoko Miyata[1], Kenichi Harada[4], Yamato Kikkawa[5], Masaki Ohmuraya[6], Kimi Araki[7], Shinya Tsurusaki[1,8], Hitoshi Okochi[1], Nobuhito Goda[2], Atsushi Miyajima[3], Minoru Tanaka[1,8]*

[1]Department of Regenerative Medicine, Research Institute, National Center for Global Health and Medicine, Tokyo, Japan; [2]Department of Life Science and Medical Bioscience, Graduate School of Advanced Science and Engineering, Waseda University, Tokyo, Japan; [3]Laboratory of Cell Growth and Differentiation, Institute of Molecular and Cellular Biosciences, The University of Tokyo, Tokyo, Japan; [4]Department of Human Pathology, Kanazawa University Graduate School of Medicine, Kanazawa, Japan; [5]Department of Clinical Biochemistry, Tokyo University of Pharmacy and Life Sciences, Tokyo, Japan; [6]Department of Genetics, Hyogo College of Medicine, Hyogo, Japan; [7]Institute of Resource Development and Analysis, Kumamoto University, Kumamoto, Japan; [8]Laboratory of Stem Cell Regulation, Institute of Molecular and Cellular Biosciences, The University of Tokyo, Tokyo, Japan

*For correspondence:
m-tanaka@ri.ncgm.go.jp

Competing interests: The authors declare that no competing interests exist.

**Abstract** Under chronic or severe liver injury, liver progenitor cells (LPCs) of biliary origin are known to expand and contribute to the regeneration of hepatocytes and cholangiocytes. This regeneration process is called ductular reaction (DR), which is accompanied by dynamic remodeling of biliary tissue. Although the DR shows apparently distinct mode of biliary extension depending on the type of liver injury, the key regulatory mechanism remains poorly understood. Here, we show that Lutheran (Lu)/Basal cell adhesion molecule (BCAM) regulates the morphogenesis of DR depending on liver disease models. Lu+ and Lu- biliary cells isolated from injured liver exhibit opposite phenotypes in cell motility and duct formation capacities in vitro. By overexpression of Lu, Lu- biliary cells acquire the phenotype of Lu+ biliary cells. Lu-deficient mice showed severe defects in DR. Our findings reveal a critical role of Lu in the control of phenotypic heterogeneity of DR in distinct liver disease models.
DOI: https://doi.org/10.7554/eLife.36572.001

## Introduction

The liver is known to possess high capacity for regeneration upon injury. In acutely injured or surgically resected livers, regeneration is usually achieved by proliferation and hypertrophy of residual hepatocytes (*Fausto and Campbell, 2003*; *Miyaoka et al., 2012*). By contrast, under chronic or severe liver injury that impairs the proliferation of hepatocytes, liver progenitor cell (LPC) has been postulated to contribute to liver regeneration by differentiating into hepatocytes and biliary epithelial cells (BECs), also known as cholangiocytes (*Thorgeirsson, 1996*; *Fausto, 2004*; *Miyajima et al., 2014*). This response is known as ductular reaction (DR), in which LPC/biliary cell with BEC marker

**eLife digest** Bile is a green to yellow liquid that the body uses to break down and digest fatty molecules. The substance is produced by the liver, and then it is collected and transported to the small bowel by a series of tubes known as the bile duct.

When the liver is damaged, the 'biliary' cells that line the duct orchestrate the repair of the organ. In fact, the duct often reorganizes itself differently depending on the type of disease the liver is experiencing. For example, the biliary cells can form thin tube-like structures that deeply invade liver tissues, or they can grow into several robust pipes near the existing bile duct. However, it remains largely unknown which protein – or proteins – drive these different types of remodeling.

Miura et al. find that, in mice, the biliary cells which invade an injured liver have a large amount of a protein called Lutheran at their surface, but that the cells that form robust ducts do not. This protein helps a cell attach to its surroundings. In addition, the biliary cells can adopt different types of repairing behaviors depending on the amount of Lutheran in their environment.

Further experiments show that it is difficult for genetically modified mice without the protein to reshape their bile duct after liver injury. Finally, Miura et al. also detect Lutheran in the remodeling livers of patients with liver disease. Taken together, these results suggest that Lutheran plays an important role in tailoring the repairing roles of the biliary cells to a particular disease. The next step would be to clarify how different liver conditions coordinate the amount of Lutheran in biliary cells to create the right type of remodeling.

DOI: https://doi.org/10.7554/eLife.36572.002

expression proliferates from the portal areas of injured livers, forming pseudo-ductular structures. DRs are frequently observed in human chronic liver diseases and rodent models including fatty liver disease and cholangiopathy (*Shafritz and Dabeva, 2002*; *Roskams et al., 2003*; *Gouw et al., 2011*; *Wood et al., 2014*). In zebrafish models, biliary cells have been reported to contribute to regenerating hepatocytes after substantial loss of hepatocytes (*Choi et al., 2014*). In mouse models, accumulating evidence by in vitro assay or transplantation experiments of biliary cells supports the presence of potential LPC with clonogenicity and bi-lineage differentiation capacity in the biliary compartment (*Suzuki et al., 2008a*; *Okabe et al., 2009*; *Dorrell et al., 2011*; *Lu et al., 2015*). In addition, a recent study using in vivo genetic lineage tracing experiment demonstrated that biliary cells can regenerate hepatocytes as facultative LPC under impaired hepatocyte regeneration in mice (*Raven et al., 2017*). Thus, DR is considered as a process of liver regeneration in chronically injured liver. In fact, genetically manipulated mice with defects in DR have been reported to show impaired recovery from chronic liver injury (*Ishikawa et al., 2012*; *Takase et al., 2013*; *Shin et al., 2015*).

To explore the nature of LPC in DR, several mouse injury models have been developed previously. In particular, two dietary models using 3,5-Diethoxycarbonyl-1,4-dihydrocollidine supplemented (DDC) diet and choline-deficient, ethionine-supplemented (CDE) diet have been extensively utilized to characterize LPC in mice for many years (*Preisegger et al., 1999*; *Akhurst et al., 2001*). Although both models induce massive DR, the pathological features resulting from these two methods are quite distinct; CDE-induced injury is thought to be a mouse model of non-alcoholic fatty liver disease with extensive hepatic damage (*Knight et al., 2005*; *Aharoni-Simon et al., 2011*), while DDC-induced injury is considered as a model of chronic cholangiopathy with portal biliary damage and severe cholestasis (*Fickert et al., 2007*). Considering the different pathological features of these two models, DR is assumed to be regulated depending on the severity and type of liver injury. In fact, it has been reported that morphological and functional heterogeneity of ductal cells is evident in DRs originating from many different pathological conditions in human patients and rodent models (*Sell, 1998*; *Alvaro et al., 2007*; *Priester et al., 2010*; *Kaneko et al., 2015*). Similar to such previous observations, CDE- and DDC-induced DR exhibited quite distinct morphology; the former extends outwards away from the portal vein, showing primitive ductules with spindle-like shape, while the latter remains around the portal vein, forming obvious bile duct-like structures. However, very little is known about the molecular mechanisms accounting for the phenotypic difference of DR among liver disease models.

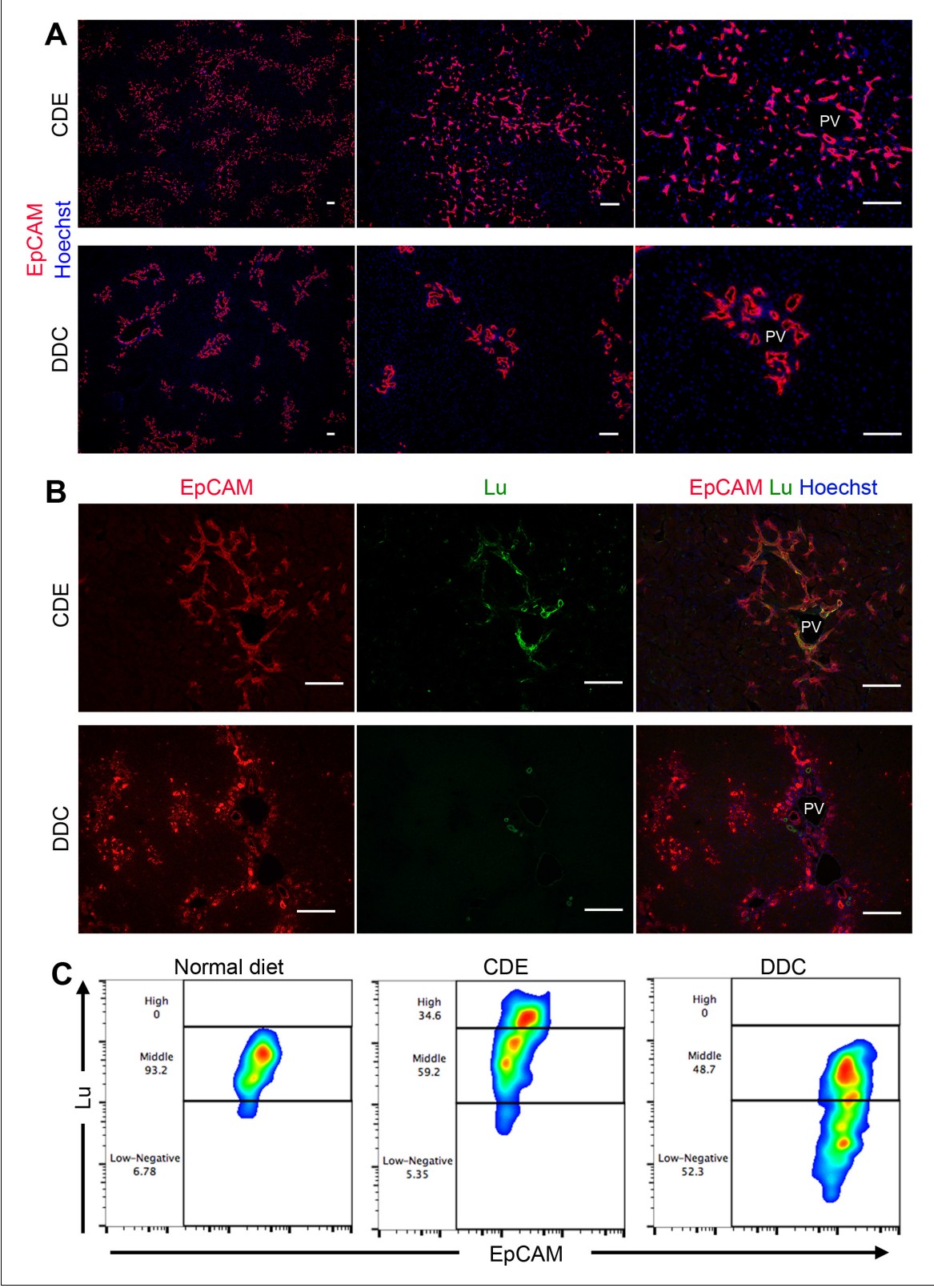

**Figure 1.** Phenotypic difference of DR and biliary cells between the CDE model and DDC model. (**A**) Immunohistochemical analysis of CDE-fed and DDC-fed mouse liver sections for EpCAM. (**B**) Co-localization of EpCAM and Lu in CDE-fed and DDC-fed mouse liver sections. (**C**) Comparison of Lu expression level in EpCAM$^+$ cells among normal diet-fed, CDE-fed and DDC-fed mouse livers by flow cytometric analysis. PV: portal vein. Scale bar: 100 μm.

*Figure 1 continued on next page*

*Figure 1 continued*

DOI: https://doi.org/10.7554/eLife.36572.003

The following figure supplements are available for figure 1:

**Figure supplement 1.** Immunohistochemical analysis of CDE-fed and DDC-fed mouse liver sections with anti-CK19 antibody.

DOI: https://doi.org/10.7554/eLife.36572.004

**Figure supplement 2.** Expression analysis for PECAM and Lu in normal and injured liver.

DOI: https://doi.org/10.7554/eLife.36572.005

**Figure supplement 3.** Co-staining of EpCAM and Lu in liver sections of normal liver.

DOI: https://doi.org/10.7554/eLife.36572.006

**Figure supplement 4.** Validation of specific reactivity of the used antibody to Lutheran.

DOI: https://doi.org/10.7554/eLife.36572.007

In the present study, we identified Lutheran blood group glycoprotein (Lu) as a crucial molecule to control the morphological heterogeneity of DR. Lu, also known as Basal Cell Adhesion Molecule (BCAM) or CD239 is a member of the immunoglobulin superfamily, which is composed of five Ig-like domains on the extracellular site, a single transmembrane domain and a short C-terminal cytoplasmic tail (*Parsons et al., 1995*). Lu is known as a laminin receptor, which has been studied in context of sickle cell disease (*Udani et al., 1998*; *El Nemer et al., 1999*), and is capable of binding to laminin-511/521 via laminin alpha5 (Lama5) chain (*Parsons et al., 2001*). It has been reported that Lu/BCAM is widely expressed in cells and tissues including hematopoietic cells, placenta and kidney, and is developmentally regulated in human liver (*Parsons et al., 1995*). However, the expression profile of Lu in mouse LPCs remains to be investigated. Here, we show that Lu is a robust marker to discriminate between CDE-induced and DDC-induced DR; Lu is highly expressed in proliferating biliary cells in the CDE model, but downregulated in the DDC model. By using fluorescence activated cell sorting (FACS), we have isolated Lu-positive and Lu-negative biliary cells from injured liver. By comparison of both biliary cells in scratch assay and cyst formation assay, we revealed a role for Lu in regulating the morphological heterogeneity of DR. Further analysis using Lu-deficient mice demonstrated that the extension of biliary tree was significantly suppressed during DR in the CDE-induced liver injury. Collectively, our findings demonstrate that Lu functionally regulates the remodeling of biliary tissue during liver regeneration, and provide new insights into the heterogeneity of LPCs among liver disease models.

## Results

### Lutheran is differentially expressed in CDE- and DDC-induced DR

We have previously identified Epithelial cell adhesion molecule (EpCAM) as a marker for murine LPC/BEC (*Okabe et al., 2009*). To investigate the difference in DRs between the CDE and DDC models, immunohistochemical analysis of EpCAM was performed. Both protocols of feeding CDE and DDC diet for 3 weeks induced robust DR accompanied by biliary cell expansion. However, the appearance of propagating ducts exhibited strikingly distinct features between two models; CDE-induced biliary cells displayed primitive capillary-like morphology, invading into the parenchymal area extensively, while DDC-induced biliary cells exhibited remarkable duct-like structure, remaining around the periportal area (*Figure 1A*). Similarly, immunostaining of CK19, another LPC marker, showed a pattern similar to that of EpCAM (*Figure 1—figure supplement 1*). Because most known LPC markers are uniformly expressed in both types of DR, these molecules may not account for the heterogeneity of LPC. Therefore, we examined the expression profile of Lu in both injury models, because we have identified Lu as a marker for hepatoblasts, a fetal type LPC during liver development. Co-staining of liver sections using anti-EpCAM and anti-Lu antibodies revealed that Lu was detected in extending biliary cells of the CDE-fed liver, whereas we could not find such signals in biliary cells of DDC-fed liver except intense signal in EpCAM⁻ ducts (*Figure 1B*). As reported by the previous paper that Lu is stained in hepatic arteries and portal vein of adult human liver (*Parsons et al., 1995*), co-staining of Lu and platelet endothelial cell adhesion molecule (PECAM), an endothelial marker, in the DDC-fed liver revealed that the EpCAM⁻ duct with strong fluorescence is hepatic artery (*Figure 1—figure supplement 2*). Next, to compare the expression level of Lu in

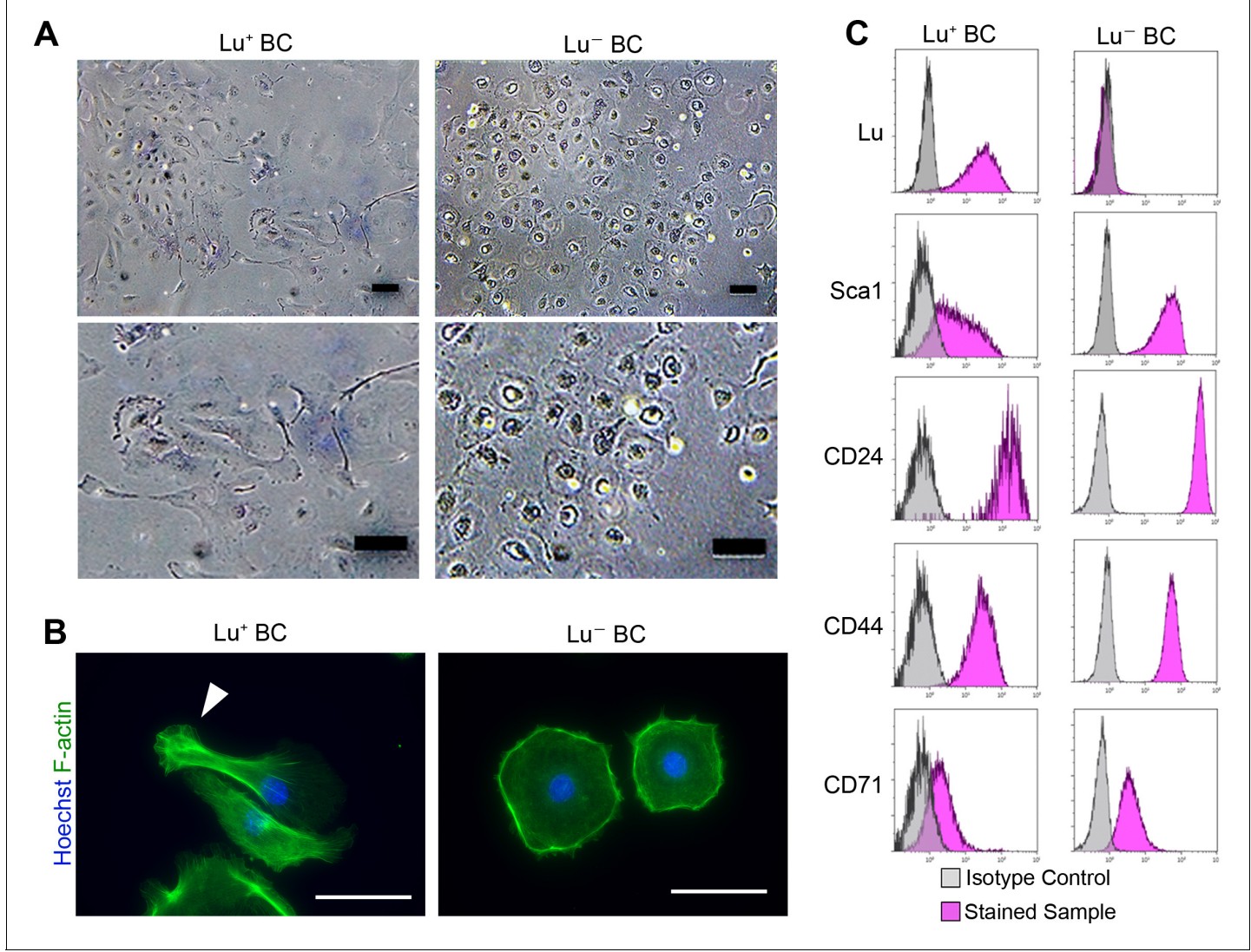

**Figure 2.** Culture of Lu+and Lu- BC isolated from injured liver. (A) Representative images of Lu+ BC and Lu- BC by bright field microscopy. (B) Immunocytochemistry for F-actin in cultured Lu+ BC and Lu- BC. Arrowhead indicates pseudopod. (C) Flow cytometric analysis of Lu, Sca1, CD24, CD44, and CD71 expression levels in Lu+ BC and Lu- BC. The cultured Lu+ BC and Lu- BC were used for analysis after 6 passages. Scale bar: 100 μm.

DOI: https://doi.org/10.7554/eLife.36572.008

The following figure supplements are available for figure 2:

**Figure supplement 1.** Expression profile of EpCAM, Lu and PECAM in non-parenchymal cells (NPCs) prepared from CDE-injured livers.

DOI: https://doi.org/10.7554/eLife.36572.009

**Figure supplement 2.** Immunocytochemistry for F-actin in cultured EpCAM+cells isolated from normal liver.

DOI: https://doi.org/10.7554/eLife.36572.010

biliary cells, we performed flow cytometric (FCM) analysis using anti-EpCAM and anti-Lu antibodies. In untreated normal liver, FCM and immunohistochemical analyses demonstrated that most EpCAM+ biliary cells showed moderate expression of Lu (*Figure 1C* and *Figure 1—figure supplement 3*). By contrast, approximately 35% of EpCAM+ cells in CDE-fed liver showed high expression of Lu, whereas nearly half of EpCAM+ cells in DDC-fed liver exhibited low or negative expression of Lu (*Figure 1C*). These opposing expression profiles of Lu in the CDE and DDC models led us to hypothesize that Lu might regulate the morphological heterogeneity of expanding biliary cells between these distinct injury models.

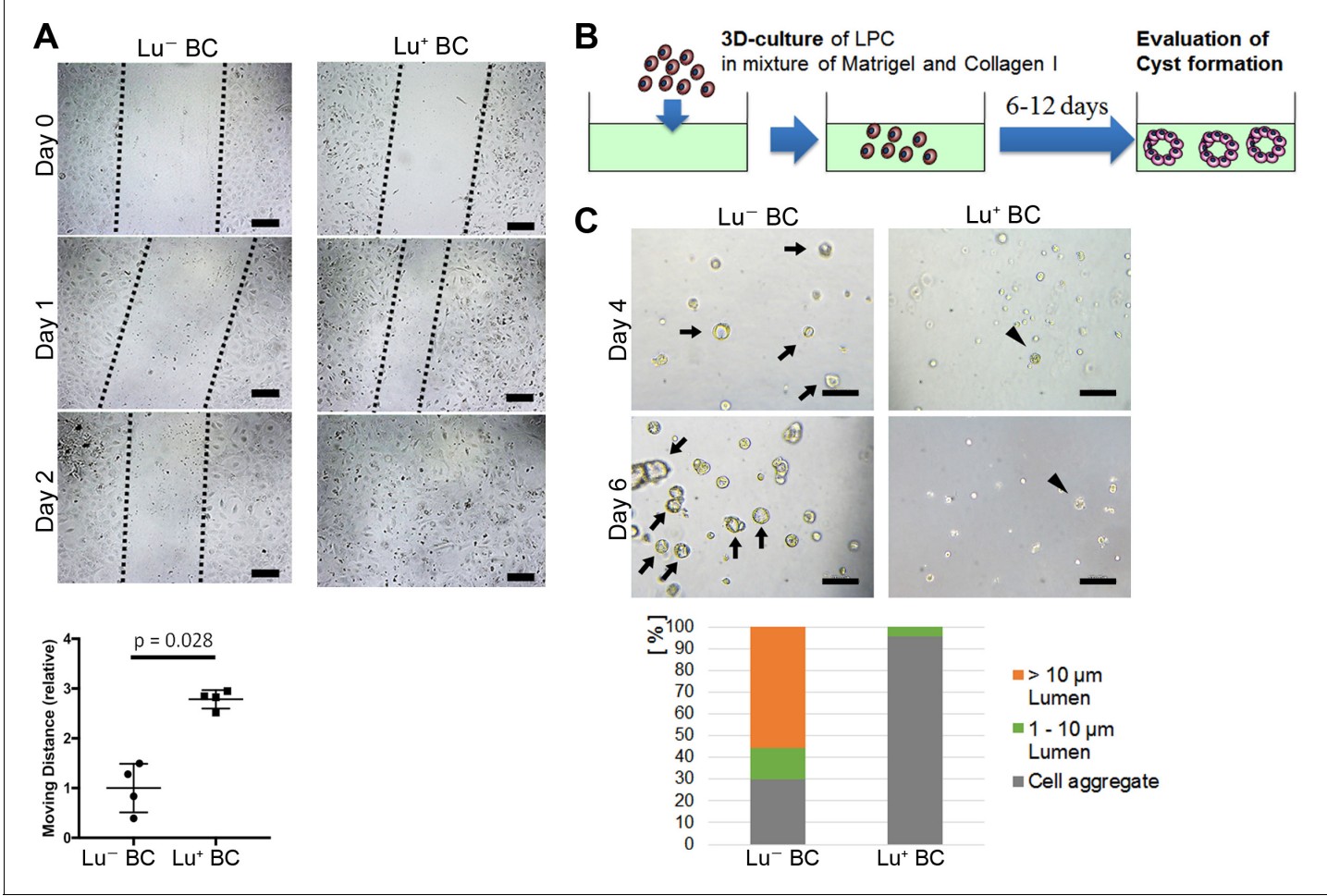

**Figure 3.** Evaluation of Lu⁻ BC and Lu⁺ BC characteristics by scratch assay and cyst formation assay. (**A**) Scratch assay using Lu⁻ BC or Lu⁺ BC. Representative images of day 0, day 1, and day 2 after scratch are shown. Quantitative data of cell moving distance at day 1 are plotted in a graph with mean and standard deviation. n = 4 biological replicates. Dotted line indicates cell front of scratched gap. (**B**) Schematic diagram of three-dimensional culture and cyst generation. (**C**) Bright field microscopic image of three-dimensionally cultured Lu⁺ BC and Lu⁻ BC at culture day 4 or day 6. Arrows and arrowheads point to cysts and cell aggregates, respectively. The details of formed cysts are shown below. The cell cluster devoid of luminal structure was regarded as 'Cell aggregate'. Scale bar: 100 μm.

DOI: https://doi.org/10.7554/eLife.36572.011

The following source data is available for figure 3:

**Source data 1.** *Figure 3A*: Numerical data for measurements of migrating distance in the scratch assay using Lu⁻ BC and Lu⁺ BC.

DOI: https://doi.org/10.7554/eLife.36572.012

## Isolation of biliary cells with a distinct level of Lu from chronically injured liver

To uncover the role of Lu in regulating biliary cell morphology, we isolated Lu^high and Lu^{-/low} biliary cell fractions from CDE-injured livers by FACS, and then cultured each population. Although Lu is highly expressed in endothelial cells, we confirmed that a contamination of such cells in EpCAM⁺ gating is highly unlikely as shown in *Figure 2—figure supplement 1*. We refer to these cultured biliary cells as Lu⁺ BC and Lu⁻ BC in the following studies. Interestingly, similar to the morphological characteristics of DR in vivo, Lu⁺ BC displayed spindle-like shape while Lu⁻ BC showed epithelial-like morphology even after several passages (*Figure 2A*). Fluorescent staining using Phalloidin demonstrated that the actin fibers and pseudopods are formed in the atypical cell body of Lu⁺ BC while F-actin accumulates in the periphery of round-shaped cell body of Lu⁻ BC (*Figure 2B*). By contrast, when freshly isolated EpCAM⁺ cholangiocytes from untreated normal liver were cultured on the

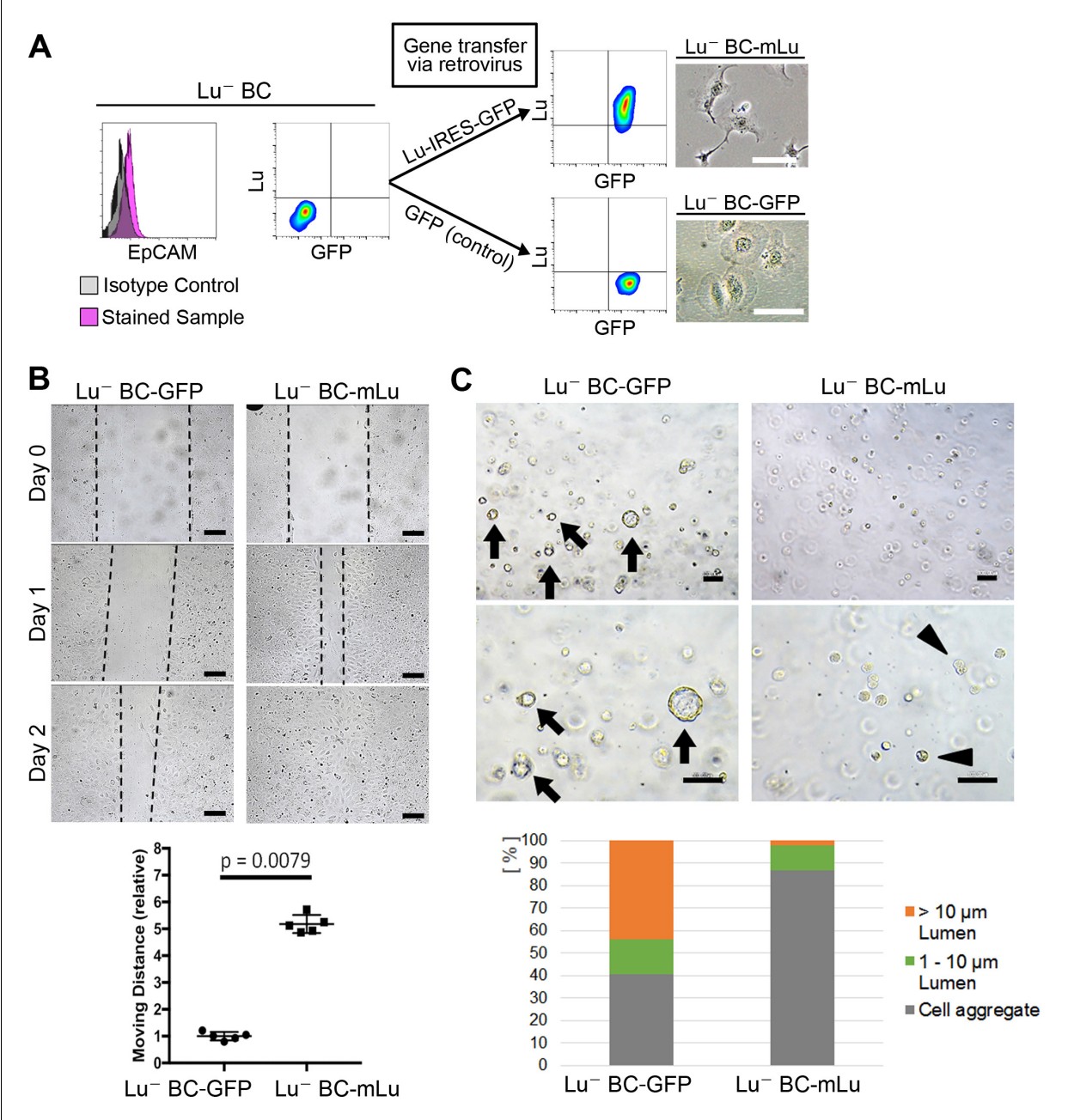

**Figure 4.** Lu regulates the motility and cyst formation capacities of biliary cells. (**A**) Schematic description for generating Lu⁻ BC-mLu or Lu⁻ BC-GFP by using retrovirus vector. Flow cytometric analyses show expression level of Lu and GFP. Bright field microscopic images show morphological change in Lu⁻ BC-mLu. (**B**) Scratch assay using Lu⁻ BC-GFP or Lu⁻ BC-mLu. Representative images of day 0, day 1, and day 2 after scratch are shown. Quantitative data of cell moving distance at day 1 are plotted in a graph with mean and standard deviation. n = 5 biological replicates. Dotted line indicates cell front of scratched gap. (**C**) Bright field microscopic image of three-dimensionally cultured Lu⁻ BC-GFP or Lu⁻ BC-mLu at culture day 6. Arrows and arrowheads point to cysts and cell aggregates, respectively. The details of formed cysts are shown below. The cell cluster devoid of luminal structure was regarded as 'Cell aggregate'. Scale bar: 100 μm.

DOI: https://doi.org/10.7554/eLife.36572.013

The following source data is available for figure 4:

**Source data 1.** *Figure 4B*: Numerical data for measurements of migrating distance in the scratch assay using Lu⁻ BC-GFP and Lu⁻ BC-mLu.
DOI: https://doi.org/10.7554/eLife.36572.014

dish, the attached cells exhibited a mixture of round and indefinite morphology (*Figure 2—figure supplement 2*). Furthermore, FCM analysis of cultured Lu$^+$ BC and Lu$^-$ BC revealed that the expression profile of Lu was maintained in each cell; Lu$^{high}$-derived biliary cells continued to express Lu at a high level, while Lu$^{-/low}$-derived biliary cells hardly expressed Lu (*Figure 2D*). By contrast, several stem cell markers such as CD24 and CD44 were expressed similarly between Lu$^+$ BC and Lu$^-$ BC, suggesting that these cells are closely related, but distinct homogeneous populations.

## Comparison of cell motility and cyst forming capacity between Lu$^+$ BC and Lu$^-$ BC

Considering that biliary cell with high expression of Lu in the CDE model exhibits an invasive phenotype in vivo, Lu$^+$ BC is expected to have a higher capacity for cell motility than Lu$^-$ BC. To compare the capacity for cell motility between Lu$^+$ BC and Lu$^-$ BC, we performed an in vitro scratch assay, by which the moving distance was evaluated after creating a scratch on the dish. As expected, Lu$^+$ BC showed significantly higher motility capacity than Lu$^-$ BC (*Figure 3A*), suggesting that there is a causal link between Lu expression level and cell motility.

On the other hand, biliary cells with low or no expression of Lu in the DDC model exhibited an obvious luminal structure around the portal vein in vivo, suggesting that Lu$^-$ BC has higher capacity for duct formation than Lu$^+$ BC. As previously reported, LPCs are able to form cyst-like luminal structures with biliary epithelial polarity in a three-dimensional (3D) organoid culture (*Tanimizu et al., 2007*). We thereby compared the cyst-forming capacity between Lu$^-$ BC and Lu$^+$ BC in the 3D culture system (*Figure 3B*). After 6 days of culture, Lu$^-$ BC formed a large number of cystic structures, while Lu$^+$ BC formed only small cell aggregates, demonstrating that Lu$^-$ BC has higher duct-forming capacity than Lu$^+$ BC (*Figure 3C*). Thus, Lu$^+$ BC and Lu$^-$ BC showed distinct features of cell motility and cyst formation in vitro. These results suggest that Lu may play a crucial role in the cell kinetics of biliary cell.

## Lu$^-$ BC acquires the characteristics of Lu$^+$ BC by overexpression of Lu

To investigate whether Lu expression is responsible for the characteristics of biliary cells, cDNA encoding either mouse Lu (mLu) and green fluorescent protein (GFP) or only GFP as a control was transduced into Lu$^-$ BC by retroviral vector. The mLu-transduced Lu$^-$ BC (Lu$^-$ BC-mLu) showed morphological change into a spindle-like shape, while GFP-transduced control (Lu$^-$ BC-GFP) retained a round shape (*Figure 4A*). In the scratch assay, Lu$^-$ BC-mLu showed higher mobility than Lu$^-$ BC-GFP (*Figure 4B*). In contrast, the cyst formation assay demonstrated the reduced cyst-forming capacity of Lu$^-$ BC-mLu, resulting in the formation of numerous small aggregates (*Figure 4C*). These results strongly suggested that Lu$^-$ BC acquired the Lu$^+$ BC-like phenotype after Lu expression, and that Lu might endow biliary cells with characteristics of spindle shape morphology and enhanced motility.

## Constitutive association of Lama5 with DR in chronically injured livers

Although the differential expression profile of Lu is likely to be relevant to the phenotypic heterogeneity of DR, the molecular mechanism by Lu remains unclear. Because Lu is known to be a receptor for Lama5 and bind to Laminin-511 and -521, we next investigated the expression of Lama5 in CDE- and DDC-injured livers. Intriguingly, double staining of EpCAM and Lama5 revealed that most expanding biliary cells are fully surrounded by Lama5 in both liver injury models (*Figure 5A*). Considering the accumulation of Lama5 in the vicinity of biliary cells, it is plausible that Lama5 may be secreted from biliary cell itself rather than the environmental niche cell. Indeed, the expression of *Lama5* mRNA was verified in both EpCAM$^+$ biliary cells isolated from CDE- and DDC-injured livers (*Figure 5B*), implying the involvement of Laminins in Lu-driven regulation. While Lu is capable of binding to Laminin-511/521 via Lama5, these laminins are also known as a ligand for Integrinα3β1/α6β1 (*Kikkawa et al., 2007*). It has been reported that Lu binds to Lama5 competitively with Integrinα3β1/α6β1 and promotes tumor cell migration by modulating Integrin-mediated cell attachment to Laminin-511 protein (*Kikkawa et al., 2013*). Taking these evidences into account, Lu may regulate the morphogenesis of DR via an Integrin-mediated manner. Given that Lu plays a role in the competitive inhibition against Laminin-511/521 and Integrinα3β1/6β1 axis in biliary cell as shown in *Figure 5—figure supplement 1*, high expression of Lu would be reproduced by inhibition of integrinβ1 (Itgb1) signaling. To address this possibility, we first investigated the expression of *Integrinα3*

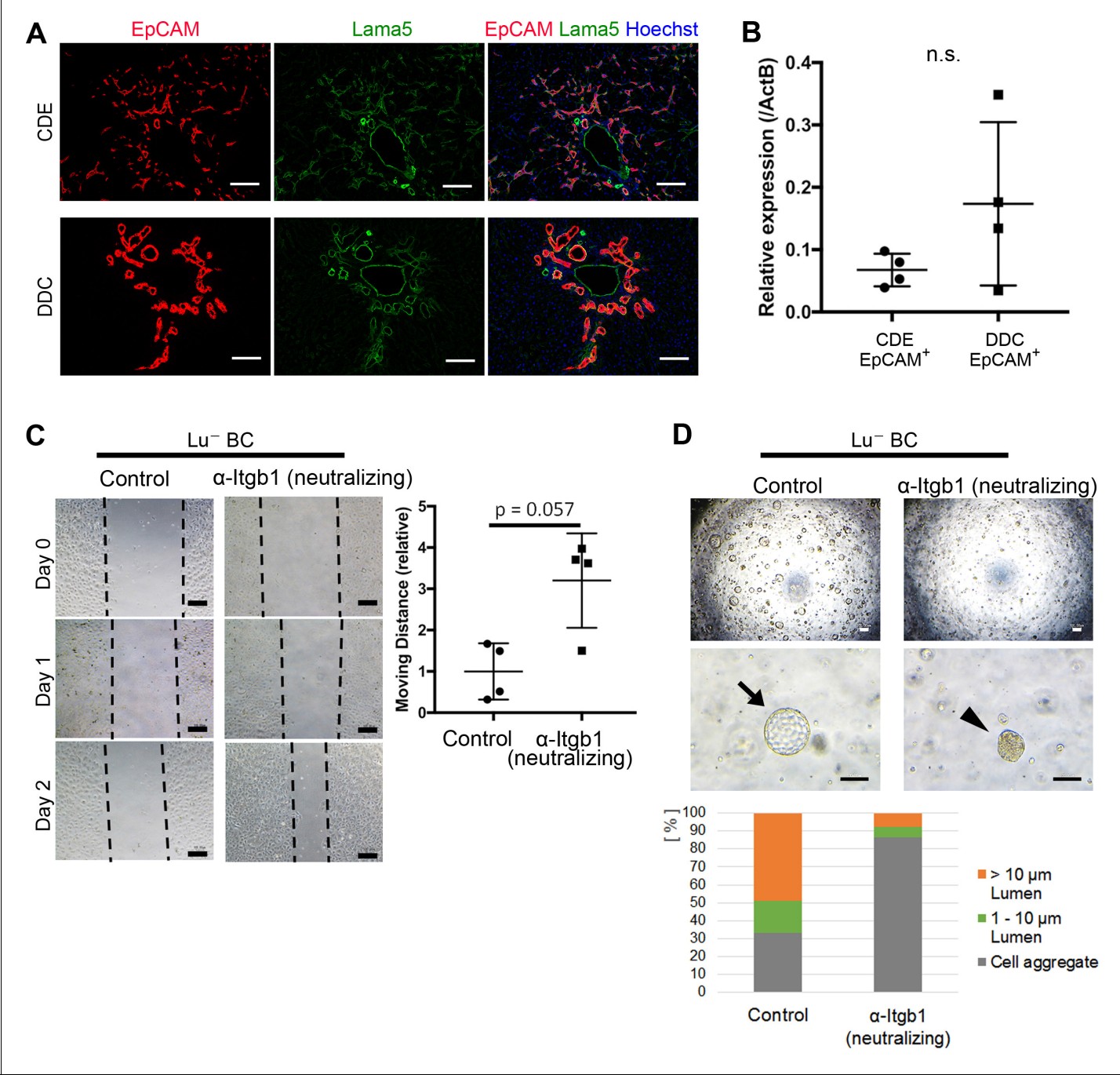

**Figure 5.** Itgb1 signaling is critical for regulating the phenotype of biliary cells. (**A**) Expression analysis for Lama5 in injured liver. Co-staining of EpCAM and Lama5 was performed in liver sections of CDE-fed mouse and DDC-fed mouse. (**B**) Evaluation of *Lama5* gene expression in EpCAM+ cells isolated from CDE-fed and DDC-fed mouse livers by quantitative RT-PCR. Data are plotted in a graph with mean and standard deviation. n = 4 biological replicates. n.s.: not significance. (**C**) Scratch assay using Lu- BC in the presence or absence of Itgb1 neutralizing antibody. Hamster IgM was used as control. Representative images of day 0, day 1, and day 2 after scratch are shown. Dotted line indicates cell front of scratched gap. Quantitative data of cell moving distance at day 1 are plotted in a graph with mean and standard deviation. n = 4 biological replicates. (**D**) Bright field microscopic image of three-dimensionally cultured Lu- BC-GFP at culture day 12 in the presence of Itgb1 neutralizing antibody or control IgM. Arrows and arrowheads point to cysts and cell aggregates, respectively. The details of formed cysts are shown below. The cell cluster devoid of luminal structure was regarded as 'Cell aggregate'. Scale bar: 100 µm.

DOI: https://doi.org/10.7554/eLife.36572.015

The following source data and figure supplements are available for figure 5:

*Figure 5 continued on next page*

*Figure 5 continued*

**Source data 1.** *Figure 5B*: Numerical data for expression analysis of *Lama5* mRNA in EpCAM⁺ cells by quantitative RT-PCR.
DOI: https://doi.org/10.7554/eLife.36572.023
**Figure supplement 1.** Schematic model of the inhibitory effect of Lutheran on Integrin signaling.
DOI: https://doi.org/10.7554/eLife.36572.016
**Figure supplement 2.** Gene expression analysis of *Itga3*, *Itga6* and *Itgb1* mRNA in Lu⁻ and Lu⁺ BC by quantitative RT-PCR.
DOI: https://doi.org/10.7554/eLife.36572.017
**Figure supplement 2—source data 1.** Numerical data for expression analysis of *Itga3, Itga6 and Itgb1* mRNA in Lu⁺ BC and Lu⁻ BC by quantitative RT-PCR.
DOI: https://doi.org/10.7554/eLife.36572.018
**Figure supplement 3.** Gene expression analysis of *Bcam* mRNA in Lu⁻ BC.
DOI: https://doi.org/10.7554/eLife.36572.019
**Figure supplement 3—source data 1.** Numerical data for expression analysis of *Bcam* mRNA in Lu⁻ BC by quantitative RT-PCR.
DOI: https://doi.org/10.7554/eLife.36572.020
**Figure supplement 4.** Cyst formation assay of Lu⁺ BC in the presence of activating anti-Itgb1 antibody (TS2/16) or control IgM.
DOI: https://doi.org/10.7554/eLife.36572.021
**Figure supplement 4—source data 1.** Numerical data for the details of formed cyst in the 3D culture using Lu⁺ BC in the presence or absence of activating anti-Itgb1 antibody (TS2/16).
DOI: https://doi.org/10.7554/eLife.36572.022

(*Itga3*), *Integrinα6* (*Itga6*) and *Itgb1* in Lu⁻ BC and Lu⁺ BC. As shown in *Figure 5—figure supplement 2*, all integrin components were expressed in Lu⁻ BC and Lu⁺ BC, indicating that Lu⁻ BC and Lu⁺ BC are potentially competent to cell signaling via Integrinα3β1/α6β1-Laminin-511/521 axis. We next examined the effect of neutralizing antibody against Itgb1 on the motility and duct formation capacity of Lu⁻ BC in vitro. Although the inhibition of Itgb1 signaling did not affect the expression of Lu (*Figure 5—figure supplement 3*), it dramatically changed Lu⁻ BC to Lu⁺ BC-like phenotype in both scratch assay and cyst formation assay (*Figure 5C and D*). Conversely, we investigated the effect of Itgb1 activation on Lu⁺ BC. Because TS2/16 antibody has been reported to activate Itgb1 signaling (*Rozo et al., 2016*), we added it to the 3D culture of Lu⁺ BC. As a result, Lu⁺ BC acquired cyst formation capacity by the activation of Itgb1 (*Figure 5—figure supplement 4*). These data strongly suggested that Lu regulates the characteristic of DR by modulating the Itgb1 signaling.

### *Bcam* KO mice show a key role for Lu in DR in CDE-induced liver injury

To verify the role of Lu/Bcam in DR in vivo, we generated *Bcam* knockout (KO) mice, in which the 11 bp deletion within a signal sequence-coding region of exon1 resulted in the frame shift of the *Bcam* gene (*Figure 6A* and *Figure 6—figure supplement 1*). The *Bcam* KO mice were healthy, showing no obvious developmental abnormality as reported previously (*Rahuel et al., 2008*). To compare the phenotype of DR between Wild-type (WT) and *Bcam* KO mice, mice were fed a CDE or DDC diet for 3 weeks and then their livers were analyzed by immunohistochemistry for EpCAM. When mice were fed the CDE diet, the proliferation of biliary cells occurred in both WT and *Bcam* KO mouse livers (*Figure 6B*). The loss of Lu protein in KO mouse was confirmed by immunostaining for Lu (*Figure 6C*). To our surprise, Lu-deficient biliary cells failed to spread outwards from the portal vein while WT biliary cells extended into the parenchymal area. The migrating distance of biliary cells from the portal vein was significantly shorter in *Bcam* KO mice than WT mice (*Figure 6D*).

By contrast, in the DDC model, the migrating distance of biliary cells from the portal vein showed no significant difference between WT and *Bcam* KO mice (*Figure 6D and E*). However, a slight hyperplasia of duct-like structures was observed in *Bcam* KO mice in both DDC and CDE models (*Figure 6F*). Moreover, freshly isolated EpCAM⁺ biliary cells from WT CDE-fed mouse livers showed spindle-like shape and pseudopods on the dish, while those from *Bcam* KO mice predominantly exhibited rounder morphology resembling Lu⁻ BC (*Figure 6G*). These results suggested that Lu plays a critical role in the definition of morphological heterogeneity of DR in vivo.

### CD239 is expressed in DR in human liver disease

To investigate the expression of Lu/BCAM (CD239) in human liver disease, we performed immunostaining of CD239 for a few resected samples obtained from the surgery to remove liver cancer.

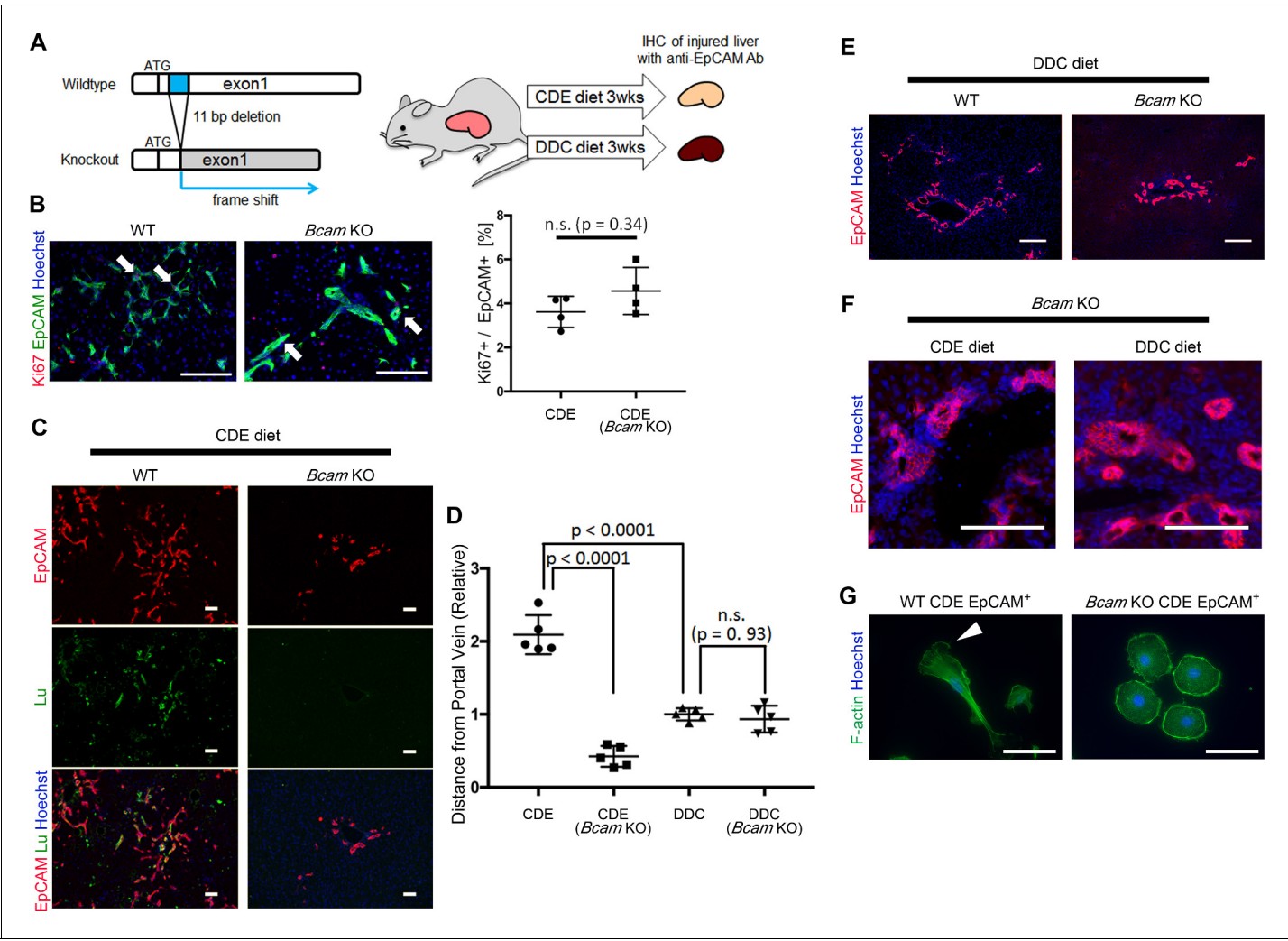

**Figure 6.** *Bcam* KO mice show drastic phenotype change in DR. (**A**) Schematic diagram of experiments using *Bcam* KO mouse. (**B**) Co-staining of EpCAM and Ki67 in WT and *Bcam* KO mouse liver sections in the CDE model. The ratio of Ki67$^+$ cell per EpCAM$^+$ cell is plotted in a graph with mean and standard deviation. n = 4 biological replicates. n.s.: not significance. (**C**) Co-staining of EpCAM and Lu in CDE-fed WT and *Bcam* KO mouse liver sections. (**D**) Quantitative analysis for the distance from portal vein to distal biliary cells in the CDE and DDC models. Data are plotted in a graph with mean and standard deviation. Statistical significance among groups is determined using one-way ANOVA. n = 5 biological replicates. (**E**) Immunostaining of EpCAM in WT and *Bcam* KO mouse liver sections in the DDC model. (**F**) Magnified immunohistochemical image of DR in CDE-fed and DDC-fed *Bcam* KO mouse liver. (**G**) Morphological image of EpCAM$^+$ cells sorted from WT and *Bcam* KO mouse fed a CDE diet for 3 weeks. The cells were directly stained with Phalloidin to visualize F-actin 72 hr after plating on culture dish. Arrowhead points to pseudopod. Scale bar: 100 μm.
DOI: https://doi.org/10.7554/eLife.36572.024

The following source data and figure supplement are available for figure 6:

**Source data 1.** *Figure 6B*: Numerical data for the ratio of Ki67$^+$ cells per EpCAM$^+$ cells.
DOI: https://doi.org/10.7554/eLife.36572.026

**Figure supplement 1.** Generation of *Bcam* KO mouse by using the CRISPR/Cas9 method.
DOI: https://doi.org/10.7554/eLife.36572.025

Intriguingly, CD239 was stained in DR in patients of chronic liver disease including non-alcoholic steatohepatitis, hepatitis B virus and hepatitis C virus (*Figure 7*), suggesting that the expression profile of Lu is conserved in humans.

## Discussion

DR is often observed in various situations of chronic liver injury or submassive liver cell loss. A number of anatomical and histological analyses of human and rodent liver tissues have supported the notion that DR represents the expansion of LPC for supplying transit-amplifying cells to replenish the damaged hepatic cells. However, the origin and the role of LPC in liver regeneration is still under intensive debate. Recent lineage tracing experiments in mice have revealed that the hepatocytic differentiation of LPCs derived from the biliary compartment is negligible in DDC-induced liver injury (*Malato et al., 2011*; *Español-Suñer et al., 2012*; *Tarlow et al., 2014*; *Rodrigo-Torres et al., 2014*). This is reasonable because DDC-injury is a model of chronic cholangiopathy, which requires replenishment of cholangiocytes or bile ducts to be recovered. In fact, it has been reported that mice with impaired DR causes severe jaundice in the DDC model (*Takase et al., 2013*). Consistently, we observed many bile ducts with an obvious luminal structure in the DR of DDC-fed liver. Therefore, the downregulation of Lu in LPC may represent a process of cholangiocytic differentiation and reinforcement of duct formation for bile excretion in the DDC model.

In contrast to the DDC-model, it is still controversial whether LPC may contribute to hepatocyte regeneration in other chronic liver injury models. The contribution of LPC derived from biliary component to hepatocyte regeneration has been reported in the CDE model using two different lineage tracing approaches based on BEC marker genes, *Osteopontin* (*Spp1*) and *Hnf1β* (*Español-Suñer et al., 2012*; *Rodrigo-Torres et al., 2014*), although to a much lesser extent. The valid but low contribution of LPC of biliary origin to hepatocytic differentiation may be explained by robust proliferation of mature hepatocytes, because hepatocyte-mediated regeneration is not inhibited in the CDE model. This notion is strongly supported by a more recent report from Forbes's group suggesting that a combination of CDE-injury and inhibition of hepatocyte proliferation causes physiologically significant and higher contribution of biliary cells to hepatic regeneration (*Raven et al., 2017*). Therefore, Lu-mediated high motility of LPC in the CDE model may contribute to the rapid delivery of hepatic progenitor cells to the damaged parenchymal area far from the portal vein.

In the DR, environmental factors play crucial roles in the regulation of LPC proliferation and differentiation. Several immune cell-derived cytokines such as TNF-related WEAK inducer of apoptosis (TWEAK), interleukin-6 and interleukin-22 have been shown to be pro-mitotic for LPCs (*Jakubowski et al., 2005*; *Yeoh et al., 2007*; *Feng et al., 2012*). Cell signaling pathways including Wnt, Notch, HGF and EGF are reportedly responsible for fate decisions of LPCs (*Boulter et al., 2012*; *Fiorotto et al., 2013*; *Kitade et al., 2013*). Although the regulatory mechanism of Lu expression in LPCs is unclear, these signaling pathways may be worthy of investigating.

Extracellular matrix (ECM) is also known to serve a niche for LPC regulation in chronic liver injury as well as liver development. Of note, laminin is an important component of LPC niche affecting cell

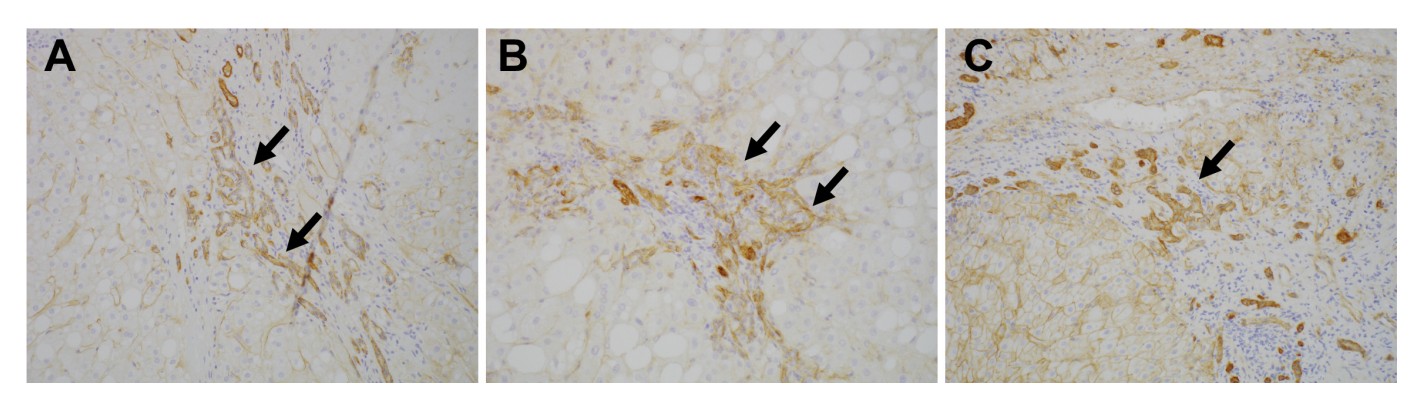

**Figure 7.** Expression of CD239 in human liver disease. Immunohistochemical images for Lu/BCAM (CD239) in human liver disease are shown. Cirrhotic liver sections obtained from the patients of non-alcoholic steatohepatitis (**A**), hepatitis B virus (**B**) and hepatitis C virus (**C**) were stained. DRs are denoted with arrows.

DOI: https://doi.org/10.7554/eLife.36572.027

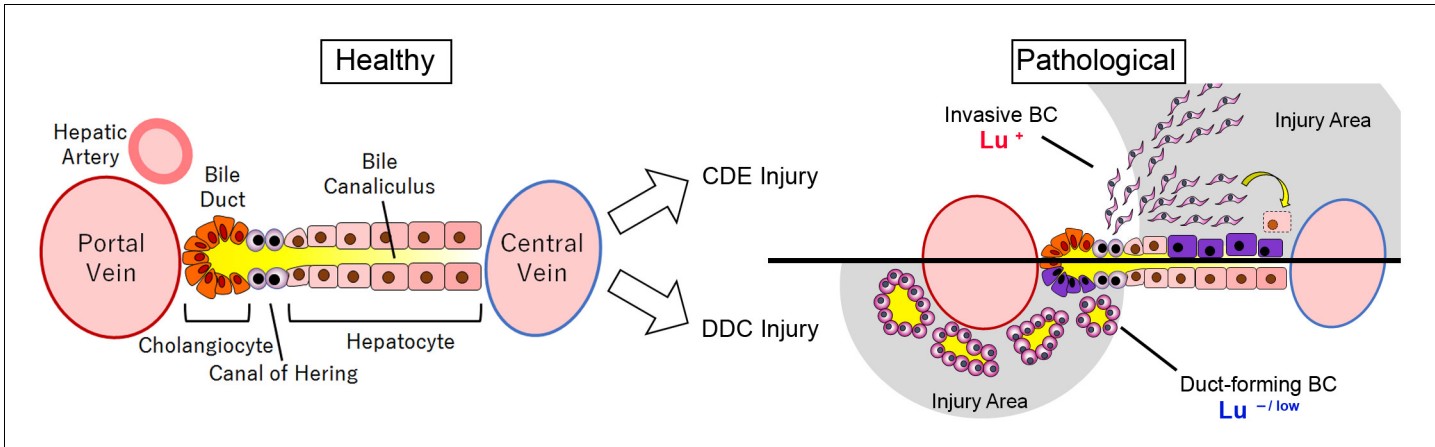

**Figure 8.** Graphical abstract of this study.
DOI: https://doi.org/10.7554/eLife.36572.028

fate decision. It has been reported that the escape of LPCs from the laminin basement favors their hepatocytic differentiation (*Español-Suñer et al., 2012*; *Paku et al., 2004*), while laminin aids maintenance of LPC and biliary cell phenotype (*Lorenzini et al., 2010*; *Boulter et al., 2013*). Consistently, it has been reported that Lama5 KO mice show defects in bile duct formation during liver development and that Itgb1 signaling is required for cholangiocytic differentiation from hepatoblast, a fetal-type LPC (*Tanimizu et al., 2012*). It is therefore highly probable that Itgb1 signaling in the context of laminin/integrin axis is crucial for biliary regeneration and duct formation from LPC. Our in vitro data also supported the idea that Itgb1 signaling modulated by Lu expression may govern DR, depending on liver injury type.

It remains undetermined whether high expression of Lu in biliary cells is an essential step for hepatocytic differentiation. Alternatively, the up-regulation of Lu may provide LPC with a cue to escape from ECM-rich periportal area, as evidenced by *Bcam* KO phenotype in the CDE model. We showed that the downregulation of Lu in ductular reactive cells facilitated the cystogenic phenotype. However, the loss of Lu would be also expected for the hepatocytic differentiation from LPC because mature hepatocytes do not express Lu. In line with the idea, intriguingly, the expression of Lu seemed to be downregulated in EpCAM[+] cells at the tip of DR in the CDE model (*Figure 1B*). The discrepancy may be explained by the microenvironment surrounding LPCs; the periportal region is rich in ECM including laminin whereas the parenchymal region is poor in it. Taking such microenvironment into consideration, the loss of Lu in parenchymal area may be a sign of hepatocytic differentiation by the escape of LPC from laminin deposition. Further expression analysis of Lu and Laminin with a combination of lineage tracing experiment will uncover the mechanism underlying LPC-mediated liver regeneration depending on the microenvironment.

In addition to the signaling molecules and ECM, several non-parenchymal cells have been shown to be involved in the microenvironment for LPC. Intriguingly, Hul et al. reported that prolonged Kupffer cell (KC) depletion did not influence the proliferation of LPCs but reduced their invasive behavior in the CDE model (*Van Hul et al., 2011*). The LPCs of KC-depleted mice exhibit phenotype resembling cells of biliary lineage with rounder morphology. More interestingly, the LPCs remain closer to the portal area, in places delineating a pseudo-lumen. This phenotypic change of LPCs by KC depletion closely resembles that of *Bcam* KO mice in the CDE diet. On the other hand, Boulter et al. reported that hepatic macrophage played a role in promoting LPC specification to hepatocytes by expressing Wnt3a in the CDE model (*Boulter et al., 2012*). These evidences suggest that hepatic macrophages may be a key regulator of Lu expression in a process of LPC specification. Further studies using a lineage tracing experiment in *Bcam* KO mice under various types of chronic liver injuries will provide a clue to better understanding the molecular mechanisms underlying the phenotypic heterogeneity, as well as fate determination of LPCs during liver regeneration.

In conclusion, the present study demonstrates that the expression profile of Lu in biliary cells dramatically changes during DR depending on the type of liver injury, which in turn dictates the

morphological characteristics of biliary cells such as cell motility and duct formation (*Figure 8*). This molecular mechanism would be expected to be conserved in human liver diseases. Thus, Lu is a novel marker for classification of DR and an interesting functional molecule for investigating the nature of LPCs. Our findings will provide new insights into the significance of biliary cell heterogeneity in liver regeneration.

# Materials and methods

## Key resources table

| Reagent type | Designation | Source or reference | Identifiers | Additional information |
|---|---|---|---|---|
| Antibody | anti-EpCAM (rat monoclonal) | PMID: 19429791 | | (1:100–500) |
| Antibody | anti-EpCAM (rabbit polyclonal) | Abcam | Abcam:ab71916; RRID:AB_1603782 | (1:400) |
| Antibody | anti-Lutheran (rat monoclonal) | this paper | | (1:500) |
| Antibody | anti-Sca1 (rat monoclonal) | BioLegend | BioLegend:108107; RRID:AB_313344 | (1:400) |
| Antibody | anti-CD24 (rat monoclonal) | Miltenyi Biotec | MB:130-102-731; RRID:AB_2656573 | (1:400) |
| Antibody | anti-CD71 (rat monoclonal) | Miltenyi Biotec | MB:130-109-632; RRID:AB_2659126 | (1:400) |
| Antibody | anti-CD44 (rat monoclonal) | Miltenyi Biotec | MB:130-110-117; RRID:AB_2658152 | (1:400) |
| Antibody | anti-FcR (rat monoclonal) | BioLegend | BioLegend:101320; RRID:AB_1574975 and BioLegend:101302; RRID:AB_312801 | (1:100) |
| Antibody | anti-CK19 (rabbit polyclonal) | PMID:12665558 | | (1:1000) |
| Antibody | anti-Lama5 (rabbit polyclonal) | PMID:9151674 | | (1:200) |
| Antibody | anti-Ki67 (rat monoclonal) | ThermoFisher | ThermoFisher:14-5698-80; RRID:AB_10853185 | (1:200) |
| Antibody | anti-PECAM (rat monoclonal) | BD Biosciences | BD:553373; RRID:AB_394819 | (1:100) |
| Antibody | anti-CD31 (rabbit polyclonal) | Novus | NB:NB100-2284; RRID:AB_10002513 | (1:100) |
| Antibody | anti-CD239 (rabbit monoclonal) | Abcam | Abcam:2994–1; RRID:AB_2065309 | (1:100) |
| Antibody | anti-CD29 (hamster monoclonal) | BD Biosciences | BD:555002; RRID:AB_395636 | 5 µg/mL for 3D culture |
| Antibody | anti-CD29 (mouse monoclonal) | BioLegend | Biolegend:303010; RRID:AB_314326 | 50 µg/mL for 3D culture |
| Antibody | Hamster IgM, λ1 isotype (hamster monoclonal) | BD Biosciences | BD:553957; RRID:AB_479639 | 5 µg/mL for 3D culture |
| Antibody | Mouse IgG1, κsotype (mouse monoclonal) | BioLegend | BioLegend:401404 | 50 µg/mL for 3D culture |
| Cell line (*Homo sapiens*) | Plat-E (Platinum-E) | PMID: 10871756 | RRID:CVCL_B488 | Cell line established in T. Kitamura lab |

## Animal models

C57BL/6J mice were purchased from Clea-Japan, Inc. (Tokyo, Japan). All animals were maintained in a standard Specific-Pathogen-Free (SPF) room at the institutional animal facility. All animal experiments were performed according to institutional guidelines and approved by the Animal Care and Use committee of the Institute of Molecular and Cellular Biosciences, The University of Tokyo (approval numbers 2501, 2501–1, 2609, 2706, and 3004), Kumatomo University (approval number A27-092), Hyogo College of Medicine (approval number 16–043, 16–046), and National Center for Global Health and Medicine Research Institute (approval numbers 15080, 16023, 17086 and 18069). To induce liver injury, a diet containing 0.1% DDC (Clea-Japan Inc. Tokyo, Japan) or the choline-deficient, ethionine-supplemented diet (MP Biomedicals, CA, USA) was fed to 6-week-old mice for 3 weeks.

## Study approval for human samples

The study using human samples was approved by the Kanazawa University Ethics Committee (approval number 305–4), and all of the analyzed samples are derived from patients who provided informed written consent for the use of their tissue samples in research.

## Antibodies

The information about antibodies used for FACS and immunohistochemistry is described in Key resources table. The rat anti-EpCAM monoclonal antibody was generated as described previously (*Okabe et al., 2009*). The rabbit anti-CK19 polyclonal antibody was generated as described previously (*Tanimizu et al., 2003*). The rat anti-Lutheran monoclonal antibody used in this study was generated by immunization of a rat with mouse fetal hepatic cells as described previously (*Suzuki et al., 2008b*), and biotinylated for FACS using ECL Protein Biotinylation Module (GE Healthcare UK Ltd, UK). The specific reactivity against mouse Lu was validated by flow cytometric (FCM) analysis of Ba/ F3 cells transfected with *Bcam* cDNA by a retroviral vector, pMxs/IRES-GFP (*Kitamura et al., 2003*) (*Figure 1—figure supplement 4*). The anti-Lutheran monoclonal antibody (D295-3) and anti-EpCAM monoclonal antibody (D269-3) are commercially available from MBL International Corporation, MA, USA.

## Liver perfusion and cell sorting

Cells were isolated from murine livers as described previously (*Okabe et al., 2009*). Briefly, liver cells were dissociated by perfusion of collagenase solution. Non-parenchymal cells (NPCs) were prepared by removal of hepatocytes with repeated centrifugation at 100 g for 2 min. Then, NPCs were incubated with anti-FcR antibody for blocking non-specific binding, followed by with fluorescein isothiocyanate (FITC)-conjugated anti-EpCAM monoclonal antibody for 30 min on ice. After incubation with anti-FITC microbeads (1:10–100 dilution, Miltenyi Biotec, Bergisch Gladbach, Germany), EpCAM$^+$ cells were enriched by autoMACS pro (Miltenyi Biotec). After MACS, cells were incubated with biotin-conjugated anti-Lutheran monoclonal antibody for 30 min on ice. After wash, cells were incubated with allophycocyanin (APC)-conjugated streptavidin (1:100–500 dilution, BD bioscience, NJ, USA) for 20 min on ice and analyzed or purified by fluorescence-activated cell sorting (FACS) using Moflo XDP (Beckman-Coulter, CA, USA) and BD FACSCanto II (BD bioscience). Dead cells were excluded by propidium iodide (Sigma-Aldrich, MO, USA) staining.

## Cell culture

Sorted EpCAM$^+$ cells from CDE-fed mouse liver were cultured with modified William's-E medium as previously reported (*Okabe et al., 2009*). For in vitro assay, the cells expanded at 3 to 6 passages were used. For cytoskeleton staining, Alexa Fluor 488 Phalloidin (1:500 dilution, Thermo Fisher Scientific, MA, USA) was used.

## Scratch assay

The cells were seeded in a 24-well plate at a confluency of 90–95% the day before scratch assay. After 12 to 24 hr of culture, the medium was removed and fresh medium with 10 µg/mL Mitomycin C (Wako Pure Chemical Industries) was added to fully confluent cells to inhibit further proliferation. The treated cells were incubated continuously for 150 min at 37°C. After incubation, the cell layer was scratched crosswise with a micropipette tip. Each well was washed twice with PBS to prevent detached cells from re-adhering. After creating the scratch, the cells were continuously cultured without cytokines. The moving distance was calculated by subtracting the half of gap length at Day 1 from that at Day 0. For experiments of antibody administration, Hamster anti-rat CD29 (555002, BD Pharmingen) or Hamster IgM (553957, BD Pharmingen) was added in the culture at a final concentration of 1 µg/mL.

## Immunohistochemistry

The resected left lobe of liver was embedded into OCT compound (Sakura Finetek Japan), and frozen by liquid nitrogen. The frozen block was cut into 8 µm slices by Microtome Cryostat HM 525 (Thermo Fisher Scientific). Fixation was performed by using 4% paraformaldehyde (Wako Pure Chemical Industries, Osaka, Japan) or cold acetone (Wako Pure Chemical Industries). For blocking buffer, 5–10% skim milk (BD bioscience) or 3% FBS (Thermo Fisher Scientific) was used. Primary antibodies used for immunohistochemistry were rabbit anti-CK19 polyclonal antibody, rat anti-EpCAM monoclonal antibody, rabbit anti-EpCAM polyclonal antibody, rabbit polyclonal anti-Laminin α5 antibody (a kind gift from Dr. Jeffrey H. Miner), rabbit anti-PECAM polyclonal antibody, rat anti-Ki67 monoclonal antibody and rat anti-Lutheran monoclonal antibody. The information about antibodies

is described in Key resources table. All images were captured using KEYENCE BZ-X710:BZ-X Viewer, Zeiss Axio observer z1: AxioCamHR3 or Olympus FV3000. The ratio of Ki67$^+$ cell per EpCAM$^+$ cell was calculated using Hybrid Cell Count function in the Dual Signal Extraction mode of BZ-X Analyzer. An average value of three random images per mouse was treated as a representative value for the mouse. Quantification of the distance of biliary cell cluster/cell from the center of the portal vein was performed using a previously reported method (*Best et al., 2016*). Briefly, the distance from the center of the portal vein to the most distal EpCAM-stained cell was measured, and then the mean diameter of the portal vein was subtracted from this value to eliminate the influence of the size of the portal vein. An average value of six to fourteen random images of portal region per mouse was treated as a representative value for the mouse.

## Three-dimensional culture

For three-dimensional culture, Cellmatrix Type I-A (Nitta Gelatin, Osaka, Japan) and Matrigel with Growth Factor Reduced (Corning, MA, USA) were used for gel components. Chilled cellmatrix and Matrigel were mixed at 1:9 ratio and used to coat the surface of the culture dish. After solidification of the coating layer by incubating at 37°C, the mixture of cell suspension and gel at 1:1 ratio was added. After 30 min of incubation at 37°C in 5% $CO_2$ chamber, culture medium was loaded on top of the double-layered gel. The top layer of medium was changed twice a week. All images were captured using DS-Fi2-L3 (Nikon Corporation, Tokyo, Japan) under a phase-contrast microscope (Nikon ECLIPSE TS100, Nikon Corporation) after 6–12 days of culture. For the experiments using neutralizing and activating antibodies against Itgb1, Hamster anti-rat CD29 (555002, BD Biosciences) and TS2/16 (303010, Biolegend) were added in the culture at a final concentration of 5 µg/mL and 50 µg/mL, respectively. For each control, Hamster IgM, λ1 isotype control (553957, BD Pharmingen) or Mouse IgG1, κ isotype control (401404, Biolegend) were used. For quantification of the size and formation efficiency of cyst, 50 cells were cultured in individual wells of a 96-well plate. After 6 days of culture, the image was captured by a phase-contrast microscope. All visible cell clusters were counted according to the diameter of lumen. The cell cluster devoid of luminal structure was counted as 'Cell aggregate'.

## Establishment of Lu-expressing Lu$^-$ BC by retroviral vector

For overexpression of mouse Lu (mLu) in Lu$^-$ BC, *Bcam* cDNA was amplified with two primers 3'- C TCGAGTCACTGCCGCCACTGCAG −5' and 3'- GTCGACTTACATTCCCTGGAGGAAG −5' by RT-PCR and inserted into the EcoRI and XhoI restriction enzyme sites of pMxs-IG plasmid vector (kindly provided by Dr. Kitamura) was used. For the production of retrovirus, pMxs-mLu-IG or pMxs-IG was transfected into the packaging cell line Platinum-E (*Morita et al., 2000*) by using lipofectamine 2000 (Invitrogen). The culture supernatant was centrifuged at 6000 g at 4°C overnight to recover virus particles. The precipitated virus particles were dissolved in culture medium and used to infect Lu$^-$ BC. After 16–24 hr, the culture media was replaced with fresh media and the culture was continued overnight. The cells expressing both mLu and GFP or only GFP were sorted by FACS and named as Lu$^-$ BC-mLu or Lu$^-$ BC-GFP, respectively.

## RNA extraction and reverse transcription

RNA extraction was performed using ISOSPIN Cell and Tissue RNA (NIPPON GENE, Toyama, Japan) according to the manufacturer's instruction. For tissue homogenization, FastPrep-24 (MP Biomedicals) was used. Reverse transcription from RNA to cDNA was performed by PrimeScript RT Master Mix (Takara-bio, Shiga, Japan).

## Quantitative RT-PCR

Quantitative RT-PCR was performed using LightCycler480 (Roche, Basel, Switzerland) with the Universal Probe Library system. The sequence of used primers is (5' to 3') EpCAM-Forward: AGAATAC TGTCATTTGCTCCAAACT, EpCAM-Reverse: GTTCTGGATCGCCCCTTC, Lama5-Forward: GGCC TGGAGTACAATGAGGT, Lama5-Reverse: CACATAGGCCACATGGAACA, ITGB1-Forward: TCAACATGGAGAACAAGACCA, ITGB1-Reverse: CCAACCACAGCTCAATCTCA, ITGA3-Forward: TCAACATGGAGAACAAGACCA, ITGA3-Reverse: CCAACCACAGCTCAATCTCA, ITGA6-Forward: GCGGCTACTTTCACTAAGGACT, and ITGA6-Reverse: TTCTTTTGTTCTACACGGACGA.

## Generation of Bcam/Lu Deficient Mice

pT7-sgRNA and pT7-hCas9 plasmid were kindly provided from Dr. Ikawa (Osaka University, Japan) (*Mashiko et al., 2013*). After digestion with EcoRI, *Bcam* mRNA synthesis was performed using an in vitro RNA transcription kit (mMESSAGE mMACHINE T7 Ultra Kit, Thermo Fisher Scientific), according to the manufacturer's instructions. A pair of oligos targeting *Bcam* gene was annealed and inserted into the BbsI site of the pT7-sgRNA vector. The sequences of the oligos were as follows: Bcam/Lu (5'- AAC CCC CTG ACG CCC GCG CA −3'), which is located at exon 1 of Bcam/Lu gene. After digestion with XbaI, gRNAs were synthesized using the MEGAshortscript Kit (Thermo Fisher Scientific). The precipitated RNA was dissolved in Opti-MEM I (Thermo Fisher Scientific) at 0.4 µg/µ L. C57BL/6N female mice (Clea-Japan Inc.) were used in this study. IVF was performed according to the Center for Animal Resources and Development's (at Kumamoto University, Japan) protocol (http://card.medic.kumamoto-u.ac.jp/card/english/sigen/manual/onlinemanual.html). Electroporated embryos were cultured in KSOM medium, and transferred the next day to the oviducts of pseudo-pregnant females on the day of vaginal plug detection. Genome Editor electroporator and LF501PT1-10 platinum plate electrode (BEX Co.Ltd., Tokyo, Japan) were used for electroporation. 50 embryos prepared were subjected to electroporation. The collected embryos cultured in KSOM medium were placed in the electrode gap filled with 5 µl of Opti-MEM I containing sgRNA and *hCas9* mRNA. The electroporation conditions were 25V, five times. The eggs were then cultured in KSOM medium at 37°C and 5% $CO_2$ in an incubator until the two-cell stage.

## Human tissue and immunohistochemistry

Three individual surgical specimens of cirrhotic liver were obtained from the patients with hepatocellular carcinoma. The deparaffinized and rehydrated sections were microwaved in EDTA buffer (pH 9.0) for 20 min in a microwave oven. Following endogenous peroxidase blocking, these sections were incubated at 4°C overnight with rabbit anti-CD239 monoclonal antibody against human Lutheran/BCAM (1:100 dilution, Epitomics, CA, USA) and then at RT for 1 hr with goat anti-rabbit immunoglobulins conjugated to peroxidase labeled-dextran polymer (K4003, Envision, Dako, Tokyo, Japan). After benzidine reaction, sections were lightly counterstained with hematoxylin.

## Statistical analysis

Statistical analyses and the determination of *p* value were performed using GraphPad Prism software. Statistical significance between two groups was evaluated using Mann-Whitney U test and considered for p<0.05. For comparison of four groups, one-way analysis of variance (ANOVA) was applied, and once F-test was significant, multiple comparisons between each group were conducted by Tukey's multiple comparisons. Statistical significance was set at two-tailed *p* values < 0.05. Values derived from at least four biological replicates were plotted in a graph with mean and standard deviation. The exact number of biological samples was described in each figure legend and source data. There was no exclusion of outliers in all experiments. Group allocation was performed without any bias. A statistical method of sample size calculation was not used during study design.

## Acknowledgements

We would like to thank Dr. Jeffrey H Miner (Washington University, St Louis, MO) for supplying anti-Laminin alpha5, and Dr. Toshio Kitamura (The University of Tokyo, Tokyo, Japan) for providing a retroviral vector, pMxs/IRES-GFP. We also thank CY Kok for her editorial assistance, Y Kamiya for mouse and technical assistance, C Koga for cell sorting, the University of Tokyo IMCB Olympus Bioimaging Center (TOBIC) for acquiring image data, and the members of the Miyajima and Okochi laboratory for their helpful discussion and suggestions. This work was supported, in part, by Grants-in-Aid for Scientific Research on Innovative Areas (26110007 to MT and 26110001 to MO) and Scientific Research A (26253023 to AM) from the Japan Society for the Promotion of Science, Japan, and by Research Program (JP17be0304201 to MT) from Japan Agency for Medical Research and Development (AMED).

# Additional information

## Funding

| Funder | Grant reference number | Author |
|---|---|---|
| Japan Society for the Promotion of Science | 26110007 | Minoru Tanaka |
| Japan Society for the Promotion of Science | 26253023 | Atsushi Miyajima |
| Japan Society for the Promotion of Science | 26110001 | Masaki Ohmuraya |
| Japan Agency for Medical Research and Development | JP17be0304201 | Minoru Tanaka |

The funders had no role in study design, data collection and interpretation, or the decision to submit the work for publication.

## Author contributions

Yasushi Miura, Conceptualization, Resources, Data curation, Formal analysis, Validation, Investigation, Visualization, Writing—original draft, Writing—review and editing; Satoshi Matsui, Data curation, Formal analysis, Validation, Investigation, Visualization, Writing—review and editing; Naoko Miyata, Formal analysis, Investigation, Data acquisition by FACS; Kenichi Harada, Resources, Data curation, Formal analysis, Validation, Investigation, Visualization, Writing—review and editing; Yamato Kikkawa, Conceptualization, Resources, Writing—review and editing; Masaki Ohmuraya, Resources, Funding acquisition, Visualization, Writing—original draft; Kimi Araki, Resources; Shinya Tsurusaki, Data curation, Investigation, Visualization; Hitoshi Okochi, Supervision, Project administration, Writing—review and editing; Nobuhito Goda, Conceptualization, Supervision, Writing—review and editing; Atsushi Miyajima, Conceptualization, Resources, Supervision, Funding acquisition, Writing—original draft, Writing—review and editing; Minoru Tanaka, Conceptualization, Resources, Supervision, Funding acquisition, Investigation, Visualization, Writing—original draft, Project administration, Writing—review and editing

## Author ORCIDs

Minoru Tanaka http://orcid.org/0000-0003-2500-7973

## Ethics

Human subjects: The study using human samples was approved by the Kanazawa University Ethics Committee (approval number 305-4), and all of the analyzed samples are derived from patients who provided informed written consent for the use of their tissue samples in research.

Animal experimentation: All animal experiments were performed according to institutional guidelines and approved by the Animal Care and Use committee of the Institute of Molecular and Cellular Biosciences, The University of Tokyo (approval numbers 2501, 2501-1, 2609, 2706 and 3004), Kumatomo University (approval number A27-092), Hyogo College of Medicine (approval number 16-043, 16-046), and National Center for Global Health and Medicine Research Institute (approval number 15080, 16023, 17086 and 18069). Every effort was made to minimize animal suffering and to reduce the number of animals used.

## Decision letter and Author response

Decision letter https://doi.org/10.7554/eLife.36572.032
Author response https://doi.org/10.7554/eLife.36572.033

# Additional files

## Supplementary files

• Transparent reporting form

DOI: https://doi.org/10.7554/eLife.36572.029

**Data availability**

All data generated or analysed during this study are included in the manuscript and supporting files. Source data files have been provided for Figures 3,4,5, 6 and Supporting figure 5.

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
