## [Decision Letter]

Thank you for submitting your article "Differential expression of Lutheran regulates biliary tissue remodeling in ductular reaction during liver regeneration" for consideration by *eLife*. Your article has been reviewed by three peer reviewers, and the evaluation has been overseen by a Reviewing Editor and Marianne Bronner as the Senior Editor. The following individual involved in review of your submission has agreed to reveal his identity: Holger Willenbring (Reviewer #3).

The reviewers have discussed the reviews with one another and the Reviewing Editor has drafted this decision to help you prepare a revised submission.

Summary:

The manuscript by Miura et al. investigates the ductular reaction, an extension of the biliary system found in most types of chronic liver injury. It has previously been shown that the morphology of ductular reaction depends on the type of liver injury, for example, preferential injury to cholangiocytes or hepatocytes. However, the mechanism underlying the different modes and morphologies of ductular reactions is not known. The authors identify Lutheran (Lu), a laminin receptor, as a regulator of ductular morphology, using mouse models in which ductular reactions are induced by feeding a diet containing CDE or DDC, which injures the hepatocytes and cholangiocytes, respectively. They show that Lu expression is high in ductular reactive cells following CDE diet and low following DDC diet, and that Lu^+^ cells are more motile but less efficient in duct formation than Lu^-^ biliary cells, using in vitro assays and new Lu knockout mice. The authors provide evidence that Lu may function by competing with integrins for binding to laminin alpha5.

Overall, the data are well presented, the figures are of very good quality and the findings are of importance, not only for the liver field. However, a number of deficiencies need to be addressed to improve the manuscript.

Essential revisions:

1) The expression of Lu needs to be better characterized.

- Lu is stated to be expressed in hepatic artery in Figure 1B but a co-staining needs to be provided to support this statement. Also, the authors should show the expression of Lu in control untreated liver by means of a EpCAM/Lu costaining.

- In the CDE model, some of the cells of the ductular reaction are Lu^+^ (high), others not. What is the topography of the Lu^+^ cells ? Lu^+^ cells indeed appear to be closer to the portal area and Lu^-^ deeper in the parenchyma. Quantification may confirm this.

- A large population of ductular reactive cells have normal Lu expression in CDE-treated mice, and half of the ductular reactive cells in DDC-treated animals do not show detectable Lu as compared to controls. Can the authors speculate on the reason for heterogeneous expression of Lu in the ductular reactive cells in CDE- and DDC-treated mice?

2) The authors propose that Lu and Integrin alpha3 beta1/alpha6 beta1 compete for binding to laminin alpha5 (Lama5), because inhibition of Integrin beta1 in cultured Lu^-^ biliary cells mimics the effects of Lu overexpression. The concept of competition between Integrin and Lu should be strengthened and additional controls are required:

- Can the authors overexpress the alpha3 (or alpha6) and beta1 integrin subunits in cultured Lu^+^ cells and verify if the cells are converted to a cystogenic phenotype?

- Is the expression of Integrin alpha3 beta1/alpha6 beta1 similar in vitro in Lu

and Lu^+^ biliary cells?

- The authors should provide information on the expression of Lu in Lu^-^ cells incubated with Integrin beta1 neutralising antibodies.

3) The expression of Lama5 raises several questions.

- Lama5 is detected in vivo around ductular reactive cells in both CDE (Lu high) and DDC (Lu low) models, yet Figure 5B suggests that Lu^-^ biliary cells express more Lama5 than Lu^+^ biliary cells. Is the difference shown in Figure 5B statistically significant?

- Macrophages play a major role in laminin deposition during liver disease, including in DDC-treated livers. The authors should compare the mRNA expression of Lama5 also in this cell type.

- In the CDE model, laminin appears to predominantly co-localise with ductular reactive cells into the parenchyma, while the Lu (high) population is restricted to a more proximal periportal zone (see comment #1). As per the authors' theory, when the cells lose Lu, laminin binding to integrin should be facilitated. Hence the cells should re-adopt a tubular, pseudo-ductular conformation, which is not the case. What is the authors' explanation?

- Stuart Forbes' group demonstrated that anchorage of liver progenitor cells to a laminin sheet maintains the biliary phenotype of the cells while loss of laminin favors hepatocyte differentiation. Also, in vivo, laminin degradation favors the differentiation of liver progenitor cells in the CDE model (PMID:22922013). Can the authors explain how they reconcile their data with the literature?

4) Lu is proposed to control cell motility and cyst formation. The experiments in Figures 3 to 5 require additional controls:

- In the DDC model, ductular reactions expands as bile duct like shape in vivo. Do the Lu^-^ biliary cells in vitro have the same shape of a control cholangiocytes? In other words, control cholangiocytes should be added as control in Figure 2.

- An additional biliary marker like EpCAM in Figure 4A would confirm that the observed changes are due to Lu expression and not to contamination with other cell types. Were these cells freshly isolated or passaged?

- Concluding that Lu^+^ cell have higher motility than Lu-/low cells requires excluding higher proliferation in Figure 3A.

- Cyst size and cyst formation efficiency should be quantified in Figures3C, 4C and 5D.

5) A number of statements need to be clarified and additional information on the methods should be provided:

- In the subsection “Isolation of biliary cells with a distinct level of Lu from chronically injured liver”, the authors state that Lu (high)-derived biliary cells continued to express Lu at high levels. Information on the number of passages in Figure 2C should be given.

- The statement "proliferating biliary cell with high expression of Lu in the CDE model" requires information about proliferation markers in Lu^+^ and Lu^-/low^ cells.

- Since Lu is also expressed in arterial branches, the authors have to describe the FACS gating strategy used to sort the biliary cells and to exclude contamination by arterial endothelial cells. Similarly, it is shown in Figure 6E that hepatocytes in the CDE model express EpCAM. How are those hepatocytes excluded from sorting ? Are those hepatocytes also Lu (high)?

6) Lutheran knockout livers treated with CDE do only show very low numbers of EpCAM^+^/HNF4^+^ cells as compared to wild-type CDE-treated livers. The authors conclude from this experiment that high expression of Lutheran "might be a process of hepatocytic differentiation of liver progenitor cells". Such conclusion is not sufficiently supported by the data, since co-expression of EpCAM and HNF4 does not provide solid evidence for hepatocyte differentiation of progenitors to hepatocytes. The reviewers suggest to remove Figure 6E which is quite preliminary.

7) It is not clear what the authors refer to by "slight dilation of duct-like structures was observed in Lu KO mice (Figure 6D and F)". Better images are required to support this statement.

8) It would considerably expand the impact of the paper if some information is provided about the expression of Lu in ductular reaction in human liver disease.

---

## [Author Response]

Essential revisions:1) The expression of Lu needs to be better characterized.- Lu is stated to be expressed in hepatic artery in Figure 1B but a co-staining needs to be provided to support this statement. Also, the authors should show the expression of Lu in control untreated liver by means of a EpCAM/Lu costaining.

Because Lu has been reported to be stained in hepatic arteries intensely and in portal vein of adult human liver (Parsons et al., 1995), we supposed that the intense signal of Lu which is EpCAM-negative corresponds to mouse hepatic arteries. To support the idea, we have performed a co-staining of Lu and PECAM (CD31), an endothelial cell marker. As expected, Lu was stained in PECAM^+^ endothelial cells including hepatic artery and portal vein. Notably, Lu was detected in hepatic artery most intensely, indicating that the Lu^+^EpCAM^-^ vessel around the portal vein is hepatic artery. We have added the immunohistochemical data in Figure 1—figure supplement 2. In addition, we performed a EpCAM/Lu costaining in control untreated liver and added the data in Figure 1—figure supplement 3 according to the reviewer’s comment. Basically, the aim of Figure 1B was to compare the expression level of Lu in ductular reaction between CDE and DDC models. Therefore, the photographic sensitivity was adjusted to the most intense fluorescence of hepatic artery without saturation, and these images were acquired at the same time of exposure on the same day. In those immunohistochemical settings, not only low/negative but also middle level of Lu in EpCAM^+^ cell shown by Figure 1C was not visible in the DDC-liver, whereas the high level of Lu in ductular reaction was detected in the CDE-liver. Thus, the undetected signal of Lu in this experimental setting does not necessarily mean no expression of Lu. Consistently, as shown in Figure1—figure supplement 3, a weak signal of Lu was detected in EpCAM+ bile duct by longer exposure while the Lu signal was saturated in hepatic artery or intensely detected in portal vein.

- In the CDE model, some of the cells of the ductular reaction are Lu^+^ (high), others not. What is the topography of the Lu^+^ cells ? Lu^+^ cells indeed appear to be closer to the portal area and Lu^-^ deeper in the parenchyma. Quantification may confirm this.

As mentioned by the reviewer, several EpCAM^+^ cells apart from the portal vein seem to be lacking in Lu signal. If it was interpreted favorably, this image may reflect the hepatocytic differentiation process of EpCAM^+^ cells at the tip of ductules. It is consistent with the lack of Lu expression in mature hepatocyte. Although the idea is very attractive, the quantification of EpCAM^+^Lu^-^ cells is difficult for some reasons mentioned below. 1) The invisible signal of Lu does not necessarily mean no Lu expression as mentioned above. Because the signal of Lu is faint even in the bile duct originally expressing Lu (Figure1—figure supplement 3), it is difficult to set the threshold of Lu^-^negative cell by immunohistochemical analysis. 2) EpCAM^+^Lu^-^ cells seem to appear as a part of ductular reaction, i.e. the tip of extending ductules. Because the tip in ductular reaction is diverse and not always apart from the portal vein in the injured liver, the quantification based on the distance from the center of portal vein may bring a misleading result. From these reasons, we think that such quantification base on immunohistochemistry should be carefully treated in terms of topography. However, the reviewer’s comment is implicative, and the expression of Lu seems to be “downregulated” apart from the portal vein. Therefore, we have added such a description about the possible topography of the Lu^+^ cells without quantification in the Discussion section (fifth paragraph). We greatly appreciate the reviewer’s suggestions.

- A large population of ductular reactive cells have normal Lu expression in CDE-treated mice, and half of the ductular reactive cells in DDC-treated animals do not show detectable Lu as compared to controls. Can the authors speculate on the reason for heterogeneous expression of Lu in the ductular reactive cells in CDE- and DDC-treated mice?

Although it is unclear whether the reviewer inquires after the mechanism of transcriptional regulation of Lu or the origin of ductular reactive cells, we speculated the reasons as follows. As shown in the FACS data on normal diet (Figure 1C), most of EpCAM^+^ cells with middle expression of Lu must be normal bile duct around the portal vein. Therefore, we speculate that approximately a half of EpCAM^+^ cells with middle expression of Lu in the CDE- and DDC-injured liver may correspond to the original bile ducts, whereas the Lu^+^ (high) or Lu^-^ cells may be derived from expandable progenitor cells. However, we have no evidence about the speculation at present, because there is no method to specifically label only ductal cells with characteristic of liver progenitor. Further study using a novel lineage tracing experiment will be required to prove the idea. Regarding the transcriptional mechanism, Lu has been originally identified as an antigen of human erythropoietic lineage, and there are consensus CACC-binding sites, a GATA-1 binding site and multiple E-boxes in the Lu promoter. Although the implication of KLF1 transcription factor in Lu expression has been previously reported in red blood cell disease, the transcriptional regulation of Lu in the other tissues remains largely unknown. We speculate that some kinds of factors implicated in each pathology, e.g. damage associated molecular patterns (DAMPs) released from damaged hepatocytes or bile acids may regulate the expression of Lu. However, we have no evidence about the opposite regulatory mechanisms of Lu expression between CDE- and DDC-injuries at present.

2) The authors propose that Lu and Integrin alpha3 beta1/alpha6 beta1 compete for binding to laminin alpha5 (Lama5), because inhibition of Integrin beta1 in cultured Lu^-^ biliary cells mimics the effects of Lu overexpression. The concept of competition between Integrin and Lu should be strengthened and additional controls are required:- Can the authors overexpress the alpha3 (or alpha6) and beta1 integrin subunits in cultured Lu^+^ cells and verify if the cells are converted to a cystogenic phenotype?

We thank the reviewer for the suggestion to strengthen the concept of competition between integrin and Lu. However, the suggested experiment may be difficult. We think that integrin signals are basically complicated due to many factors, e.g. a variety of combination patterns of alpha and beta subunits, distinct affinity to various ECM, and complex intracellular signaling including Fak and p130Cas pathways. Because we cannot control each expression level of two distinct transfected genes, the balance between alpha and beta subunits are supposed to be out of order in this experiment. In addition, it is unclear whether the transfected integrins can work well, because we could not find such experiments in the previous papers. Alternatively, it may be straightforward to show that the activation of beta1 integrin changes the phenotype of Lu^+^ cells into a cystogenic one. Therefore, we performed the addition of agonistic anit-beta1 integrin antibody to Lu^+^ cells in the 3D culture. As expected, Lu^+^ cells were converted to a cystogenic phenotype. We have added the new data in Figure 5—figure supplement 4.

*- Is the expression of Integrin alpha3 beta1/alpha6 beta1 similar* in vitro *in Lu^-^ and Lu^+^ biliary cells?*

We have compared the transcriptional level of Integrin alpha3 beta1/alpha6 beta1 between Lu^-^ and Lu^+^ biliary cells by real-time RT-PCR. As a result, both biliary cells expressed Integrin alpha3 beta1/alpha6 beta1 similarly suggesting that these cells are potentially competent to cell signaling via beta1 integrin-laminin axis. We have added the new data in Figure 5—figure supplement 2.

- The authors should provide information on the expression of Lu in Lu^-^ cells incubated with Integrin beta1 neutralising antibodies.

According to the reviewer’s comment, we investigated the expression of Lu by real-time RT-PCR. As a result, there was no change of Lu expression after the incubation with neutralizing antibody. We have added the data in Figure 5—figure supplement 3.

3) The expression of Lama5 raises several questions.- Lama5 is detected in vivo around ductular reactive cells in both CDE (Lu high) and DDC (Lu low) models, yet Figure 5B suggests that Lu^-^ biliary cells express more Lama5 than Lu^+^ biliary cells. Is the difference shown in Figure 5B statistically significant?

Although the information on statistical analysis in Figure 5B was included in a source file “Figure 5B_Mann-Whitney.xlsx” at 1^st^ submission, we forgot to mention it in the main figure. Therefore, we have specified “n.s.” in the revised Figure 5B. The scattering value of Lama5 expression in DDC samples may be due to the quality of freshly isolated EpCAM+ cells because DDC model causes cholangiopathy resulting in the damage or activation of EpCAM+ biliary cells. Considering the expression of Lama5 at a protein level in Figure 5A, we think that there is no apparent difference of Lama5 expression in most of ductular reactive cells between CDE and DDC models.

- Macrophages play a major role in laminin deposition during liver disease, including in DDC-treated livers. The authors should compare the mRNA expression of Lama5 also in this cell type.

According to the reviewer’s comment, we isolated macrophages from injured livers as CD45+Ly6G-F4/80+CD11b+ fraction by FACS, and examined mRNA expression of Lama5 in those cells by real-time RT-PCR. Compared to the EpCAM^+^ biliary cells, the expression level of Lama5 was marginal or not detected in each macrophage isolated from CDE- and DDC-injured livers as shown in Author response image 1. Although macrophages play a major role in ECM deposition during liver disease, we concluded that macrophages per se are not a producer of Lama5, but rather an activator of ECM-producing cells as reported in our paper (PMID:28779552, Matsuda et al. Hepatology 2018).

- In the CDE model, laminin appears to predominantly co-localise with ductular reactive cells into the parenchyma, while the Lu (high) population is restricted to a more proximal periportal zone (see comment #1). As per the authors' theory, when the cells lose Lu, laminin binding to integrin should be facilitated. Hence the cells should re-adopt a tubular, pseudo-ductular conformation, which is not the case. What is the authors' explanation?

We think the discrepancy may be explained by a different microenvironment surrounding the EpCAM^+^ cells between in vivo and in vitro. It is known that the extracellular matrix such as laminins and collagens are rich around the portal vein in vivo, while it is relatively poor in the parenchymal area. By using the 3D culture system that is ECM-rich condition including collagens and Matrigel, we evaluated the “cystogenic capacity” of Lu^+^ and Lu^-^ cells in vitro. However, the amount and composition of ECM would be quite distinct between at the deeper area in the parenchyma and the periportal zone in vivo. Therefore, the microenvironment in the parenchyma would be unsuitable for solid tubular formation. We think that the liberation of biliary cells from ECM-rich periportal zone into ECM-poor parenchymal area is a most important role of Lu as evidenced by Lu KO phenotype. As mentioned above, we think that the EpCAM^+^ cells losing Lu at the tip of ductular reaction may be differentiating into hepatocytic cells. If the expression of Lama5 is also downregulated in those cells, the integrin signal via Laminin-511 may be attenuated, which is not the case with Lu^-^ BC in ECM-rich periportal area of the DDC model. In either case, the morphological phenotype of ductular reaction would be controlled by not only Lu expression but also the microenvironment in vivo. We have discussed about the importance of microenvironment in the Discussion section (fifth paragraph). We believe that Lu orchestrates the complicated integrin signaling depending on the microenvironment surrounding ductular reactive cells.

*- Stuart Forbes' group demonstrated that anchorage of liver progenitor cells to a laminin sheet maintains the biliary phenotype of the cells while loss of laminin favors hepatocyte differentiation. Also,* in vivo*, laminin degradation favors the differentiation of liver progenitor cells in the CDE model (PMID:22922013). Can the authors explain how they reconcile their data with the literature?*

We basically agree with the idea, and our paper is consistent with the previous works referred to by the reviewer. As previously reported by us (Kaneko et al., 2015), ductular reactive cells substantially function as bile duct to drain bile. As shown in this paper, extending ductular reactive cells are basically surrounded by Lama5, suggesting that laminin sheet is required to maintain the biliary phenotype. By contrast, Lu seems to be downregulated in EpCAM^+^ cells at the tip of ductular reaction. If such EpCAM^+^ cells are differentiating into hepatocytic cells, the downregulated Lu may contribute to the decrease of laminin deposition surrounding those cells. We have added the speculation in the Discussion section (fifth paragraph).

4) Lu is proposed to control cell motility and cyst formation. The experiments in Figures 3 to 5 require additional controls:- In the DDC model, ductular reactions expands as bile duct like shape in vivo. Do the Lu^-^ biliary cells in vitro have the same shape of a control cholangiocytes? In other words, control cholangiocytes should be added as control in Figure 2.

We are not sure what cells the reviewer considers “control cholangiocytes” in this comment. If the reviewer considers EpCAM^+^ cells isolated from untreated normal liver as control cholangiocytes, we do not think it is a suitable control for comparison with Lu^+^ and Lu^-^ BC. In this study, Lu^+^ and Lu^-^ BCs were proliferative and used for every assay after at least 3 passages. On the other hand, as previously reported (Okabe et al., 2009), approximately 0.5% of EpCAM^+^ cells (50 cells per 10,000 cells) in normal liver can survive and proliferate to form a colony in this culture. We demonstrated that the colony forming cells include the cell having LPC-like properties. In other words, most of EpCAM^+^ cells i.e. mature cholangiocytes will be dead or excluded at a relatively early stage under this culture condition. Because the passaged EpCAM^+^ cells in the same manner do not represent normal cholangiocytes, such cells will be inappropriate for control cholangiocytes. Alternatively, we investigated the shape of EpCAM^+^ cells isolated from untreated normal liver after 7 days of primary culture. Although most of cholangiocytes were dead during the culture, the attached and survived cells showed a mixture of round and indefinite shape. Although we are not sure whether this data would be informative in this study, we have added the data in Figure 2—figure supplement 2.

- An additional biliary marker like EpCAM in Figure 4A would confirm that the observed changes are due to Lu expression and not to contamination with other cell types. Were these cells freshly isolated or passaged?

At first, the cells expanded at 3 to 6 passages were used for all in vitro assays as described in Materials and methods section (subsection “Cell Culture”). In the case of Figure 4A, we used the cells after 3 passages for transduction of Lu cDNA or GFP cDNA. We have added the FACS data about the EpCAM expression of the parental cells in Figure 4A, indicating that there is no contamination of other cell types.

- Concluding that Lu^+^ cell have higher motility than Lu-/low cells requires excluding higher proliferation in Figure 3A.

For scratch assay, we treated the tested cells with Mitomycin C to stop the proliferation as described in Materials and methods section (subsection “Scratch Assay”). Therefore, cell proliferation does not affect cell motility in this assay.

- Cyst size and cyst formation efficiency should be quantified in Figures3C, 4C and 5D.

We evaluated the cyst size and cyst formation efficiency in 3D assays. We have added the new data on quantification regarding Figures 3C, 4C, 5D and newly added Figure5—figure supplement 4.

5) A number of statements need to be clarified and additional information on the methods should be provided:- In the subsection “Isolation of biliary cells with a distinct level of Lu from chronically injured liver”, the authors state that Lu (high)-derived biliary cells continued to express Lu at high levels. Information on the number of passages in Figure 2C should be given.

The cells after 6 passages were used for FACS analysis. We have added the information about the passage number in the legend of newly assigned Figure 2D.

- The statement "proliferating biliary cell with high expression of Lu in the CDE model" requires information about proliferation markers in Lu^+^ and Lu^-/low^ cells.

Because Ki67^+^ cells were observed in the ductular reactive cells in the CDE model as shown in Figure 6B, we have described "proliferating biliary cell” unintentionally. We have no intention of claiming the correlation between cell proliferation and Lu expression at all. In fact, as shown in Figure 6B, the ratio of Ki67^+^ cells per EpCAM^+^ cells was equivalent between WT and Lu KO mice, strongly suggesting that Lu expression is not involved in the regulation of cell proliferation. Because this sentence may be misleading for now, we have deleted “proliferating” in the text.

- Since Lu is also expressed in arterial branches, the authors have to describe the FACS gating strategy used to sort the biliary cells and to exclude contamination by arterial endothelial cells. Similarly, it is shown in Figure 6E that hepatocytes in the CDE model express EpCAM. How are those hepatocytes excluded from sorting ? Are those hepatocytes also Lu (high)?

According to the reviewer’s advice, we have added the FACS gating strategy in Figure 2—figure supplement 1. For FACS, we used only non-parenchymal cell (NPC) fraction, of which the hepatocyte fraction was removed by repeated centrifugation at 100g for 2 minutes. Although the information is described in the reference (Okabe et al., 2009), we have added it in the Materials and methods section (subsection “Liver Perfusion and Cell Sorting”). In addition, we have added FACS data of triple staining with anti-EpCAM, anti-Lu and anti-PECAM antibodies in order to clarify no contamination of endothelial cells in the purified EpCAM+ cell fraction (Figure 2—figure supplement 1). These data demonstrated that a contamination of endothelial cells was highly unlikely. In addition, our culture condition is not suitable for passage of endothelial cells because it does not contain VEGF.

Regarding the Lu expression in EpCAM^+^ hepatocyte-like cells, a co-staining of the liver tissue was technically difficult because Lu staining required acetone fixation, which spoiled HNF4α staining at the same time. Because we have removed Figure 6E as described below, we did not refer to the expression of Lu in EpCAM^+^ hepatocytic cells in the text.

6) Lutheran knockout livers treated with CDE do only show very low numbers of EpCAM^+^/HNF4^+^ cells as compared to wild-type CDE-treated livers. The authors conclude from this experiment that high expression of Lutheran "might be a process of hepatocytic differentiation of liver progenitor cells". Such conclusion is not sufficiently supported by the data, since co-expression of EpCAM and HNF4 does not provide solid evidence for hepatocyte differentiation of progenitors to hepatocytes. The reviewers suggest to remove Figure 6E which is quite preliminary.

We are grateful for the reviewers’ suggestions. According to the comments, we have removed Figure 6E. We would like to focus on the morphological features of ductular reaction rather than cell fate in this paper.

7) It is not clear what the authors refer to by "slight dilation of duct-like structures was observed in Lu KO mice (Figure 6D and F)". Better images are required to support this statement.

In the use of phrase, we have mistaken “hyperplasia” for “dilation”. We have revised it and added magnified images to show the hyperplasia of duct-like structures in Figure 6F.

8) It would considerably expand the impact of the paper if some information is provided about the expression of Lu in ductular reaction in human liver disease.

With the cooperation of Prof. Kenichi Harada who is an expert on the pathology of liver diseases at Kanazawa University, we investigated a few cirrhotic liver samples obtained from the surgery of HCC. As expected, Lu was stained in ductular reaction in patients of liver diseases including NASH, HBV and HCV. We added new data in Figure 7.